# Leveraging Print Debugging to Improve Code Generation in Large Language Models

## Abstract

Large language models (LLMs) have made significant progress in code generation tasks, but their performance in tackling programming problems with complex data structures and algorithms remains suboptimal. To address this issue, we propose an in-context learning approach that guides LLMs to debug by using a "print debugging" method, which involves inserting print statements to trace and analysing logs for fixing the bug. We collect a Leetcode problem dataset and evaluate our method using the Leetcode online judging system. Experiments with GPT-4 demonstrate the effectiveness of our approach, outperforming rubber duck debugging in easy and medium-level Leetcode problems by 1.5% and 17.9%.

## 1 Introduction

The progress achieved in large language models (LLMs) has unveiled vast possibilities for their practical implementation in code generation tasks (Chen et al., 2021; Chowdhery et al., 2022). These models can now generate code that meets basic requirements. However, their performance remains suboptimal when confronted with problems necessitating intricate data structures and algorithms, such as some competition-level problems. For instance, GPT-4 achieves nearly a 76% accuracy on easy-level Leetcode problems while a mere 26% and 7% accuracy on medium-level and hard-level Leetcode problems respectively (OpenAI, 2023). Therefore, recent studies try to explore the potential of empowering LLMs to debug (Peng et al., 2023; Chen et al., 2023; Zhang et al., 2023b; Jiang et al., 2023; Olausson et al., 2023; Sakib et al., 2023; Shinn et al., 2023) for a better performance. In Reflexion (Shinn et al., 2023), researchers propose a framework that LLMs reflect on the failed test case and maintain reflective text from the subsequent trials. Self-debug (Chen et al., 2023) teaches LLMs to perform rubber duck debugging, which means debugging with the line-by-line explanation of the code. However, these methods do not provide access to real-time variable values or the ability to trace the flow of execution, which is crucial to debug with code which includes complex algorithms. Also, these methods can't make full use of the test cases, since only telling LLMs the failed test case itself is still hard to help locate the bug.

In this work, to address these issues, we use in-context learning (Brown et al., 2020) to guide LLMs to debug with a "print debugging" method, inspired by the fact that human programmers frequently employ this method due to its simplicity and effectiveness, particularly when grappling with complex algorithms. Commonly referred to as tracing in software engineering, this method involves inserting print statements into the code to output variable values, enabling the flow of data and execution to be readily traced for easier bug identification. Figure 1 presents a visual depiction of the workflow between two distinct debugging methods: rubber duck debugging (Chen et al., 2023) and our proposed print debugging. In our method, LLMs attempt to solve the problems and receive feedback from the environment (judging system). If they fail on specific test case, the "print debugging" method is employed. LLMs are instructed to first add print statements into the code, which is then executed to capture the output from these statements. Subsequently, the model identifies the bug by seeking the inconsistency between the explanation of the corresponding test case and the outputs from the added print statements[1]. Finally, the model fixes the bug based on the above analysis. This iterative debugging process continues until the generated code passes all test cases or reaches a predefined stopping criterion.

---

[1] In this paper, we sometimes use the word "log" to represent the outputs from the added print statements.

We collect a dataset of Leetcode problems from the Leetcode[2] website for a better evaluation of our method. We contend that "print debugging" is commonly employed to debug code encompassing intricate data structures and algorithms. Additionally, we submit the solutions generated by LLMs to the Leetcode online judging system[3] for testing. We consider that a commercial judging system offers a more precise testing by subjecting the code to a broader range of unreleased and comprehensive test cases, thereby mitigating the likelihood of undetected errors compared with other benchmarks (Liu et al., 2023).

We conduct experiments with GPT-4[4] from OpenAI. The results on the Leetcode problems in easy and medium level demonstrate the substantial effectiveness of our approach in facilitating bug identification and resolution. Specifically, print debugging outperforms rubber duck debugging (Chen et al., 2023) by 1.5% and 17.9% in easy-level and medium-level Leetcode problems, respectively. However, we observed that in hard-level problems, neither print debugging nor any other debugging methods yielded improvements, resulting in a mere 5% accuracy. This outcome can be attributed to the inherent complexity of hard-level Leetcode problems, which often necessitate the utilization of sophisticated algorithms, and no debugging method alone could directly address this underlying issue. We acknowledge that further research is required to explore, for instance, the incorporation of external knowledge to assist the models in addressing such challenges.

To summarize, our contributions are:

- We propose a novel approach that harnesses the capabilities of large language models to execute print debugging.

- We release a new programming problems dataset which contains latest Leetcode questions in 3 different levels: easy, medium, hard.

- We conduct extensive experiments with GPT-4 on our collected Leetcode dataset, demonstrating that our approach brings significant improvement when compared with rubber duck debugging.

## 2 RELATED WORK

**Chain-of-thought prompting** With the recent advancements in large language models, researchers have discovered that utilizing the chain-of-thought (CoT) (Wei et al., 2022) techniques can significantly improve reasoning abilities. (Wei et al., 2022) introduced the concept of few-shot CoT, which involves generating intermediate reasoning steps before arriving at the final answer with in-context demonstrations. This approach deviates from traditional few-shot prompting (also called in-context learning (Brown et al., 2020)) that directly generate the final answer. Zero-shot CoT (Kojima et al., 2022) is another method leveraging chain-of-thought which adding the prompt "Let's think step by step." after the task description to activate LLMs to generate rationales in order for improved task performance. Other researchers also propose various prompting methods to enhance model capabilities, including auto-cot (Zhang et al., 2022), least-to-more (Zhou et al., 2022), decomposing prompting (Khot et al., 2022) and tree-of-thought (Yao et al., 2023). In our work, the explanation of logs and test cases can be seen as kind of chain-of-thought, since all these explanations serve as intermediate steps for fixing bugs in the code.

**Prompting with feedback** Despite the remarkable capabilities of large language models, it can still be challenging sometimes to generate the correct answer in a single attempt. Recently, people find LLMs can receive feedback from external environment or generated by themselves and iteratively refine according to the feedback. Self-refine (Madaan et al., 2023) launches a novel approach that allows LLMs to iteratively refine outputs with the feedback without any labeled data. Reflexion (Shinn et al., 2023) proposes a "verbal reinforcement learning" that LLMs reflect on failures based on feedback and store reflexion in a text style for future trials. REMEMBERER (Zhang et al., 2023a) employs a method that let LLMs learn experience which stored in an external memory from the feedback in the training set and transfer that experience to the test set for a better performance.

---

[2]https://leetcode.com/

[3]Every problem has a URL to submit for testing. For example, the problem in Figure 1 can be submitted in https://leetcode.com/problems/find-the-prefix-common-array-of-two-arrays/.

[4]https://platform.openai.com/docs/models/gpt-4

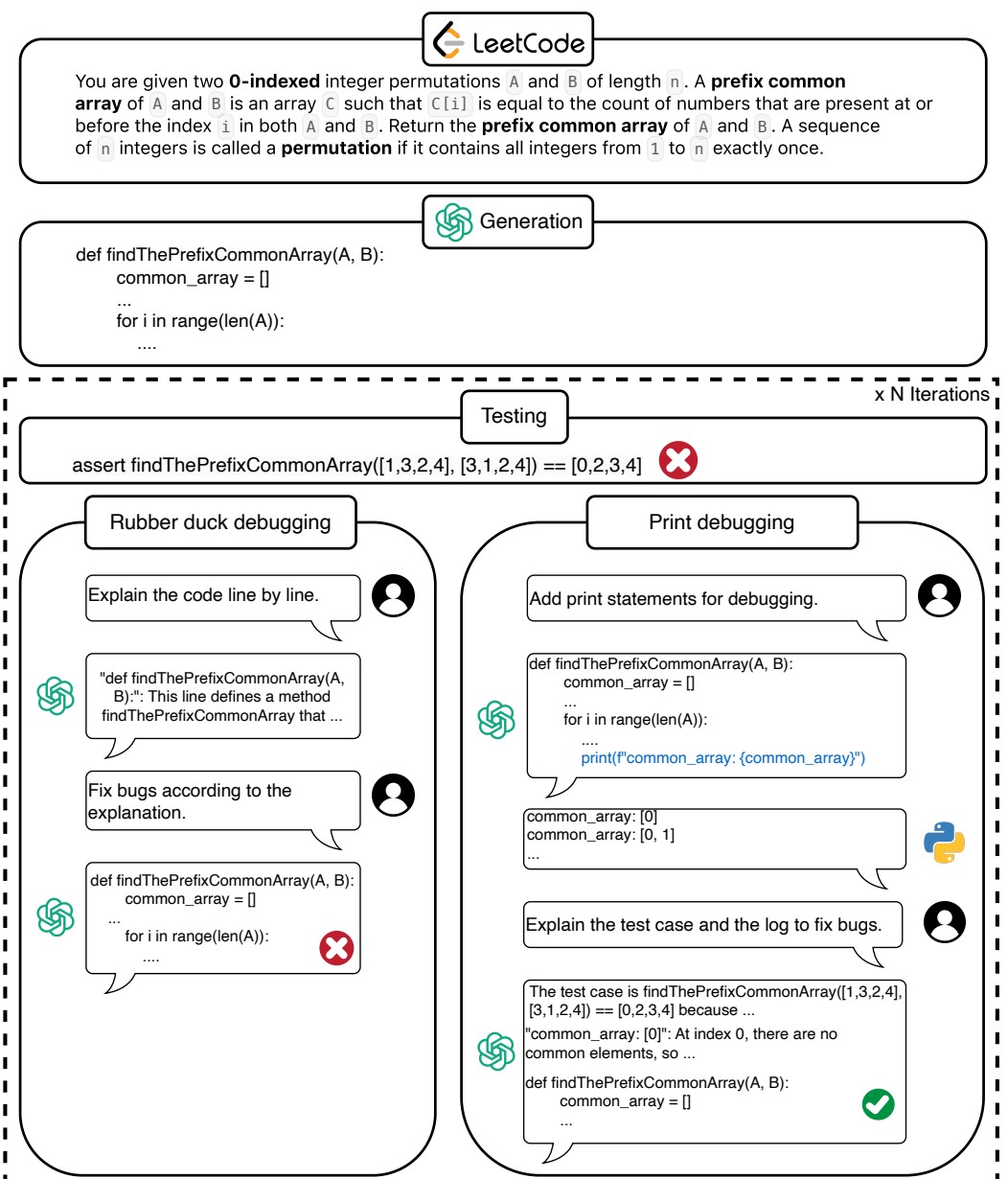

Figure 1: Comparative Workflow: Print Debugging vs. Rubber Duck Debugging. From up to down, LLMs generate code for a Leetcode problem, and subsequently subject it to testing on the Leetcode online judging system. If not all the test cases are passed, LLMs proceed to the debugging procedure. For rubber duck debugging (left down), LLMs explain the code line by line and then fix the bug according to the explanation. For print debugging (right down), LLMs insert print statements, get the log and debug according to explanation of test case and the log.

In our work, we focus on code generation task and teach LLMs to conduct print debugging, which can provide and receive much more informative feedback on the program's execution for improved debugging.

**Prompting for code** Prompting techniques have been extensively utilized in tasks related to code. Some works including Li et al. (2023c;a;b) focus on leveraging prompting to enhance code generation. (Zhang et al., 2023c) utilize prompting to facilitate code selection and develop a reviewer

model. Another line of work is to debug with LLMs, including (Peng et al., 2023; Chen et al., 2023; Zhang et al., 2023b; Jiang et al., 2023; Olausson et al., 2023; Sakib et al., 2023). The main distinction between our work and them is that we focus on using print debugging to improve performance of debugging, whereas all these previous works primarily rely on receiving execution results or error messages from the interpreter.

## 3 OUR METHODS

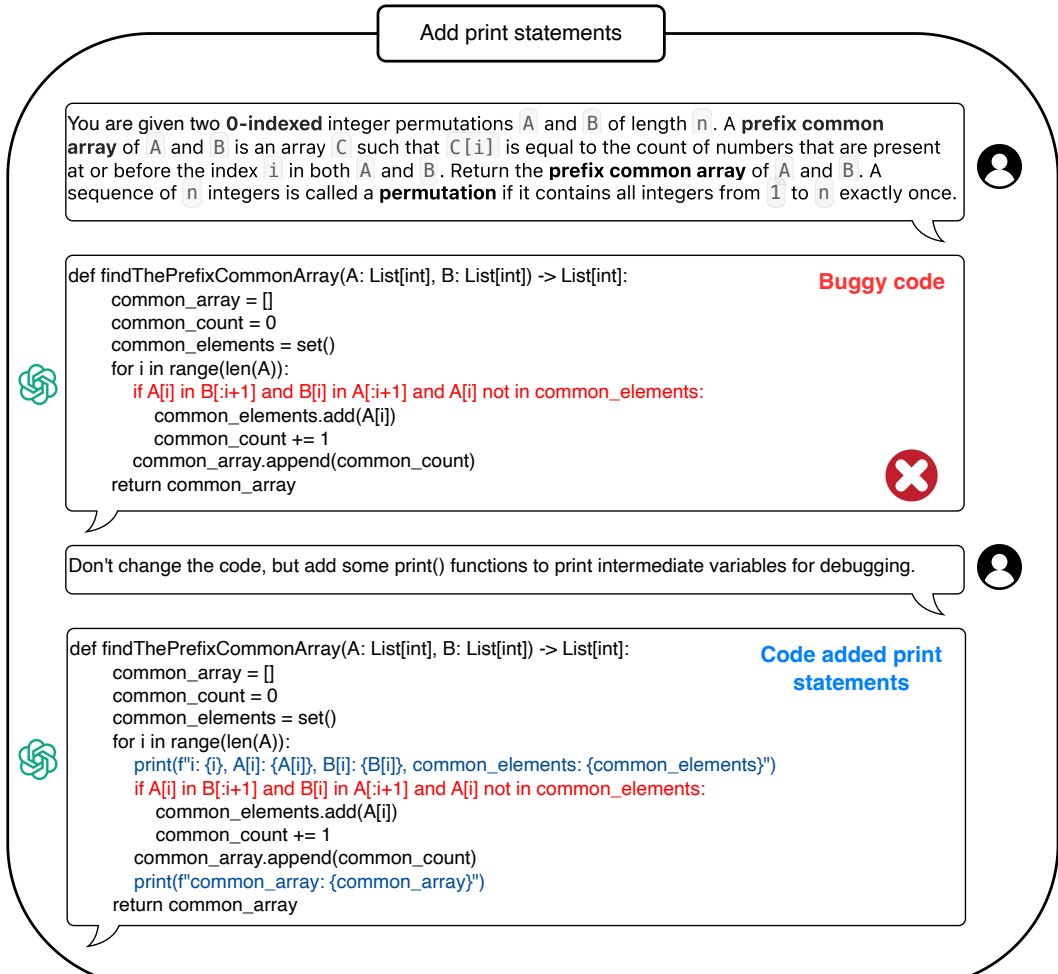

Figure 2: Illustration of adding print statements into the buggy code. LLMs are prompted to add several print statements to the buggy code, but don't change the rest of the code. Red: Buggy code, Blue: Added print statements. For the sake of brevity, we omit demonstrations and some instructions.

Our proposed method enables large language models to to employ the "print debugging" method, akin to how human programmers approach debugging. It involves inserting print statements or log messages at some strategic points in the code to output information about the program's execution flow, variable values, and other relevant data. Programmers analyse the output generated by these print statements to locate and fix bugs. This approach provides direct insight into the execution flow of a process and is particularly useful when the task involves complex data structures or algorithms. We first let LLMs attempt solving the programming problem based solely on the problem description, without any extra information. If the initial trial fails to pass all test cases, the problem enters our debugging procedure, which comprises three steps: (1) Adding print statements (2) Execution

(3) Analysing & Fixing. The above steps will be repeated until all test cases pass or until several rounds of debugging attempts still fail to fix the issues. We guide LLMs to follow these steps using one-shot prompting[5]. The complete prompts we use can be found in Appendix A. We will now provide a detailed discussion of each step:

**Adding print statements.** In this step, LLMs add print statements to the buggy code while keeping the rest of the code unchanged. Figure 2 depicts this step. A piece of code can be formalized as $S = [s_1, s_2, ..., s_i, ..., s_n]$ where each $s_i$ represents a line of code. The code with added print statements can be represented as $S_p = [p_1, s_1, p_2, s_2, ..., p_i, s_i, ..., p_n, s_n, p_{n+1}]$, where $p_i$ denotes a possible added print statement ($\emptyset$ if not added). The LLMs determine the locations and number of print statements to add by themselves. As illustrated in Figure 2, the LLMs add print statements in two places: (1) Printing the value of $i$, $A[i]$, $B[i]$ and $common\_elements$ at the beginning of the loop (2) Printing the value of $common\_array$ at the end of the loop. These print statements help capture the changing state of the common elements set and common array as the loop progresses, which is crucial for understanding how the program computes the requested prefix common array. These print statements help generate useful logs for debugging purposes, as demonstrated below.

**Execution.** In this stage, we proceed with the execution of the code that includes the added print statements from the previous step, specifically using the failed test case. During execution, we gather the output generated by the print statements. Additionally, we capture the answer or any error messages provided by the interpreter, if applicable. It is important to note that even if the execution encounters errors, we still collect the log generated prior to the occurrence of the error. For instance, in cases of array out-of-bounds errors, we can still obtain the output from the print statements preceding the termination of the process. As depicted in Figure 3, the model incorporates the output obtained from the print statements during execution as logs for subsequent debugging procedures.

**Analysing & Fixing.** In the final step, we provide the test case, the wrong answer (Chen et al., 2023) or error messages to the model and the output from the print statements. We instruct LLMs to explain the test case and the log and compare them to figure out the buggy code, as illustrated in Figure 3. For the test case explanation, the LLMs is prompted to explain it "step by step" (Kojima et al., 2022). Following the approach in (Chen et al., 2023), for the log explanation, we prompt the model to explain "line by line". LLMs are prompted to compare the explanation of both test case and the log to find the inconsistencies (Figure 3 marks one inconsistency in our showcase and see more in case study 4.5), which draws inspiration from human programmers, who often seek inconsistencies between test cases and logs to identify bugs. Subsequently, we prompt LLMs to summary and find the buggy code. Once the analysis is completed, the model is then prompted to fix the bug based on the aforementioned analysis.

## 4 EXPERIMENTS

### 4.1 SETUPS

We use `gpt-4-32k` for all our experiments. The model grants access to all the test cases within our experimental setting, which aligns with the practicality of test-driven development in software engineering (Olausson et al., 2023). We employ one-shot prompting to guide the model, repeating the debugging procedure until either all test cases were passed or 3 consecutive rounds failed to yield any improvement. In instances where debugging methods require the logs, we truncate the logs due to the limited context of GPT models. The temperature is set to 0 and the max tokens is 4096. We use accuracy as the evaluation metric, representing the percentage of problems that successfully passed all test cases.

### 4.2 COMPARED METHODS

**Simple feedback.** (Chen et al., 2023) This feedback approach solely informs the LLMs about the correctness of the submitted code, without providing any additional information.

---

[5]The complexity of the only example used in our demonstration is much lower then problems in Leetcode and only used to guide LLMs to follow the print debugging procedures and generate in the format.

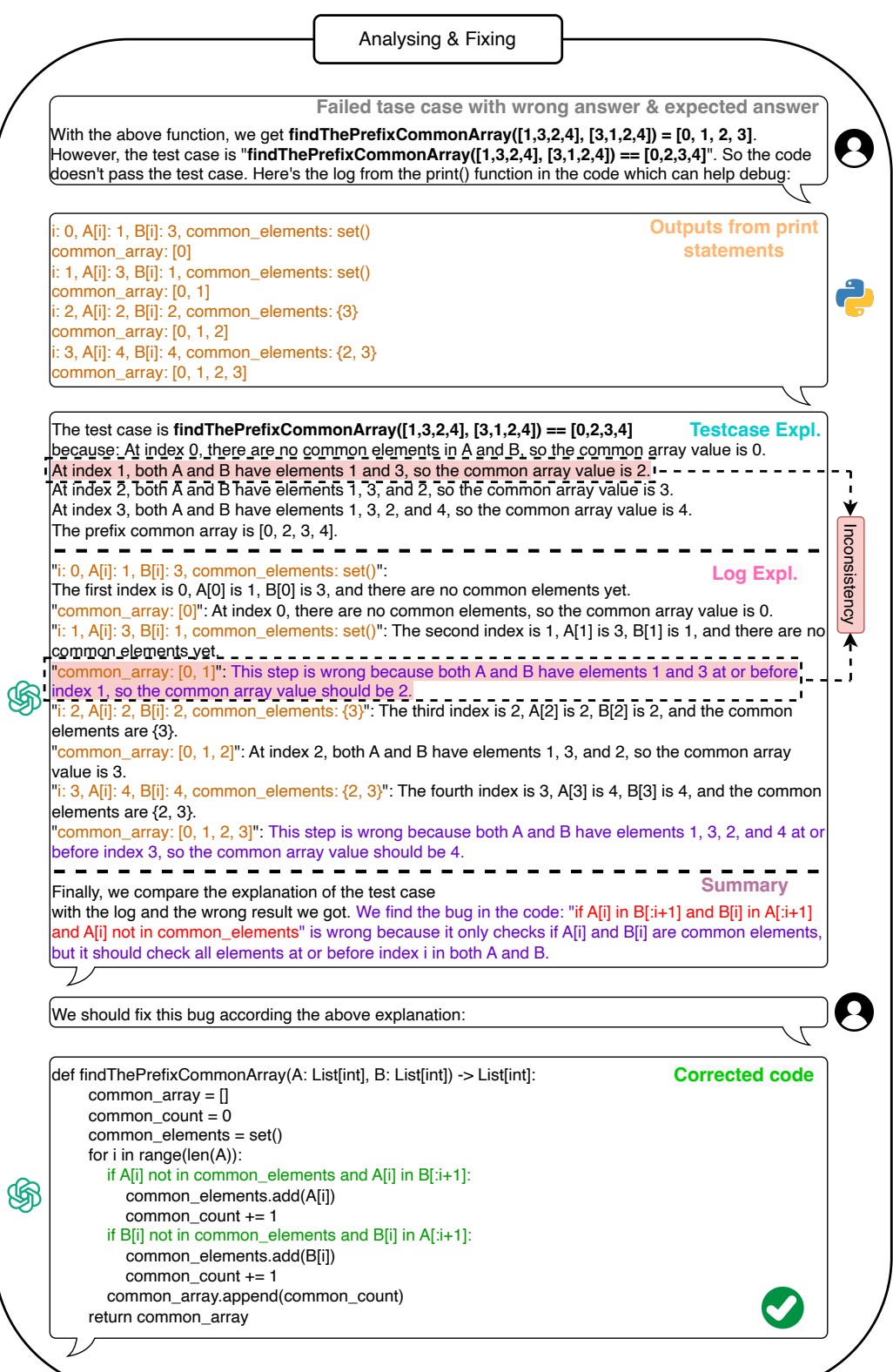

Figure 3: Illustration of analysing and fixing the bug. **Bold**: the failed test case with wrong answer and (or) expected answer, Yellow: the log, Purple: the rationales indicating the LLMs found the bug through the log, Red: the founded buggy code, which is also marked in red in Figure 2, Green: the corrected code. The inconsistencies between test case and log explanation are highlighted by background color.

Table 1: Accuracy(%) on Leetcode using GPT-4 and one-shot setting. "w/o debugging" indicates the results obtained by allowing the model to complete the code without any further debugging.

| Method | Easy | Medium | Hard |
|---|---|---|---|
| w/o debugging | 76.5 | 15.3 | 5.0 |
| simple feedback(Chen et al., 2023) | 78.8 | 20.5 | 5.0 |
| ut feedback(Chen et al., 2023) | 85.6 | 25.6 | 5.0 |
| rubber duck debugging(Chen et al., 2023) | 90.2 | 23.1 | 5.0 |
| **print debugging(ours)** | **91.7** | **41.0** | **5.0** |

**Unit test feedback.** (Chen et al., 2023) This approach returns the details of the failed test case, including the input, wrong answer, error messages if applicable, and expected answer.

**Rubber duck debugging.** (Chen et al., 2023) This method extends the information provided by unit test feedback by enabling the language model to explain the code line-by-line, aiding in the debugging process.

## 4.3 MAIN RESULTS

**Dataset.** We collect programming problems from Leetcode platform[6]. All the problems are categorized into three levels according to Leetcode platform: easy, medium, and hard. The dataset contains 132 easy problems, 39 medium problems and 40 hard problems. We use problems released after September 2019, when the GPT series model finished its pre-training[7]. Further details regarding the dataset can be found in Appendix B.

**Evaluation.** We submit the solutions generated by the model to the Leetcode platform for evaluation. The platform provided results on whether all test cases are passed. Additionally, if a solution failed, the platform returned the input, wrong answer, and expected answer of the first encountered failed test case.

**Results.** We present main results in table 1. Our method outperforms Rubber duck debugging by 1.5% and 17.9% respectively in easy and medium Leetcode problems. However, for the hard dataset, none of the debugging methods here demonstrated improvements, as all methods achieved a pass rate of only 5%. We argue that our method is more effective when dealing with problems that involve relatively complex data structures and algorithms. In the case of easy-level Leetcode problems, which primarily assess foundational knowledge of programming language usage and basic data structures, our method could not fully leverage its advantages. However, in the medium-level Leetcode problems, our method exhibited significant improvements. For the hard-level problems, we found that in most cases the model could not understand requirements or select the proper algorithms.

## 4.4 ABLATION STUDY

To further analyse the effectiveness of different components in our method, we conducted ablation studies on the easy-level and medium-level Leetcode problems. We skip studies on hard-level ones since no methods work here. Specifically, we explored the impact of (1) using only test case explanations and (2) using only the log and its explanation. The results of these ablation studies are presented in 2. Our findings indicate that removing any part of the analysis process resulted in a drop in performance and only both test cases and logs can bring improvement by helping seek inconsistencies for debugging. This highlights the importance of both test case explanations and the log in effectively debugging the code. By utilizing both sources of information, our method achieves superior results compared to when either component is omitted.

---

[6]https://leetcode.com/contest/
[7]https://platform.openai.com/docs/models/gpt-4

Table 2: Ablation studies on the easy-level and medium-level Leetcode problems. "case expl." stands for using only test case explanations and "log expl." stands for using only the log and its explanation. "all" means using all the components in our method.

| Method | Easy | Medium |
|---|---|---|
| case expl. | 89.4 | 35.9 |
| log expl. | 90.1 | 35.9 |
| **all** | **91.7** | **41.0** |

## 4.5 CASE STUDY

In this section, we demonstrate the effectiveness of our method through a case study. To minimize comprehension costs, we continue with the problem presented in Figure 1. We provide more examples in Appendix C. Our print debugging method successfully fixes the bugs in problem in Figure 1 but all other debugging methods compared fail. It's a medium-level Leetcode problem that involves finding the prefix common array of two arrays. Figures 2 and 3 together depict a round of print debugging. Due to space limitations, we only display the final round here. However, it is noted that the debugging process involves iterative attempts until a stopping criteria is met. Detailed information for all rounds is provided in Appendix D. In Figure 2, the model initially generates code containing a bug[8]. Subsequently, the model adds two print statements within the buggy code, precisely around it, enabling the tracking of data flow changes before and after executing the buggy code. Moving to Figure 3, the model first obtains information related to the test case and the output from the print statements. Then, following the instructions, the model successfully explains the test case and interprets the log. In correct code, the explanations for the test case and log should be mutually corroborative. However, in the presence of buggy code, conflicts between the test case and log arise, which we refer to as "inconsistency". In this case, the model identifies two instances of inconsistency concerning the common_array, located at index 1 and 3 (we only highlight one at index 1 for simplicity in the background color). In the subsequent summary, the model successfully identifies the code segment containing the bug based on the mentioned inconsistencies and provides an explanation for the existence of the bug. In the final correction phase, the model updates the erroneous code based on the aforementioned explanation, resulting in a corrected version that passes all test cases.

## 5 ANALYSIS

**Rounds of debugging.** First, we investigate the impact of rounds of debugging on performance. Figure 4 depicts the number of problems that succeed after debugging with different debugging methods as the number of iterations increases in the medium-level Leetcode problems. Notably, our print debugging method exhibits a continuous increase in performance, requiring a greater number of rounds (up to 7 rounds) to reach the optimal performance. In contrast, other methods tend to saturate after the initial round of debugging. These findings suggest that print debugging provides useful information to LLMs, enabling them to continuously enhance the code within several rounds of trials.

**Added print statements and generated log.** We conducted further analysis to assess the effectiveness of our print debugging method. All statistics presented herein are based on all the rounds of the debugging process for each problems in medium-level problems. Figure 5 illustrates the distribution of the number of added print statements in the code. On average, LLMs add 2.51 print statements, with the majority of cases involving the addition of 2 or 3 print statements. This practice aligns with the reasonable approach of incorporating a limited number of print statements to capture specific parts of the code during each round of debugging. Furthermore, Figure 6 showcases the distribution of the number of lines in the generated logs from the print statements. Approximately 17% of the rounds encountered log lengths that exceeded a predefined limit (commonly resulted by inserting an added print statements in a infinite loop). Consequently, such cases are not shown in the figure. On

---

[8]At this stage, the model is unaware of the specific location of the bug. The highlighted buggy code in red is derived from the model's analysis in Figure 3. We only highlight it here for clarity.

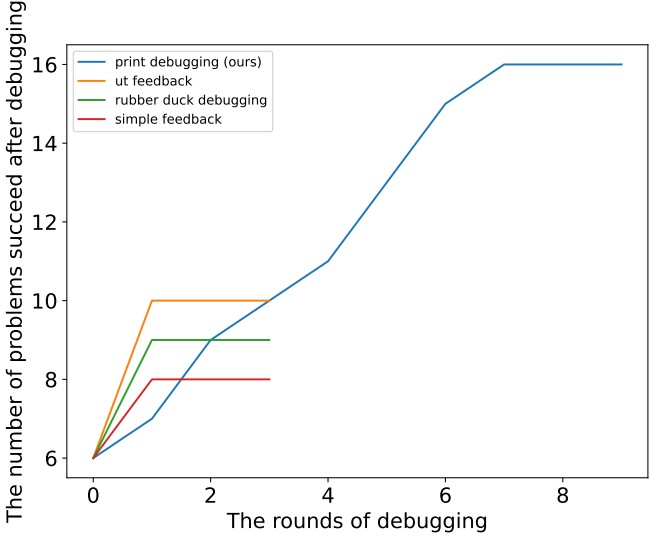

Figure 4: Performance of different debugging methods as the procedure progresses.

average, the number of lines in the logs is 11.59. The majority of logs (over 91%) comprised fewer than 20 lines, which is considered an appropriate length for current LLMs to analyse effectively.

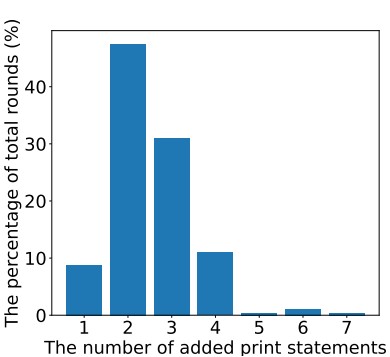

Figure 5: Number of added print statements.

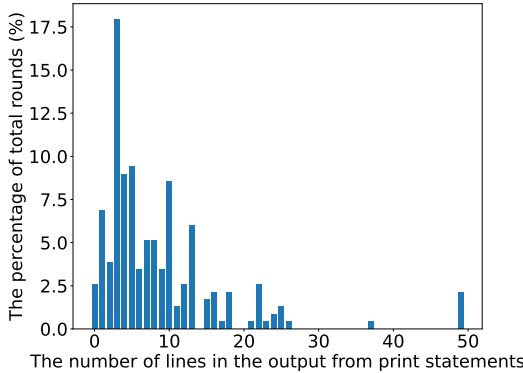

Figure 6: Number of lines in the output from print statements.

## 6  CONCLUSION

In this work, we propose an in-context learning approach to leverage LLMs to conduct a "print debugging" method, which is useful when handling programming problems with complex data structures and algorithms. We collect competition-level problems from Leetcode and submit solutions generated by LLMs to its online judging system fro evaluation. Experimental results conducted with GPT-4 demonstrated the effectiveness of our approach, surpassing the rubber duck debugging in easy and medium-level Leetcode problems by 1.5% and 17.9% respectively. This work validates the applicability of LLMs in employing print debugging methods. However, we have observed that existing LLMs still exhibit low accuracy when tackling problems that require advanced algorithms. We believe that relying solely on debugging methods is not sufficient to address these challenges. In light of future research directions, we propose the incorporation of external knowledge to tackle these issues effectively.

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

## A    FULL PROMPTS FOR PRINT DEBUGGING

```
**Task Start**
**Description Start**
You are given a list of numbers.
You need to return the sum of squared numbers in the given list, round
    each element in the list to the upper int(Ceiling) first.
class Solution:
  def sum_squares(self, lst: List[int]) -> int:
**Description End**
**Code Start**
class Solution:
  def sum_squares(self, lst: List[int]) -> int:
    sum_of_squares = 0
    for num in lst:
      rounded_num = int(num)
      sum_of_squares += rounded_num**2
    return sum_of_squares
**Code End**
Feedback: Don't change the code, but add some print() functions to print
    intermediate variables for debugging.
**Code Start**
class Solution:
  def sum_squares(self, lst: List[int]) -> int:
    sum_of_squares = 0
    for num in lst:
      rounded_num = int(num)
      print(f"num: {num}, rounded_num: {rounded_num}")
      sum_of_squares += rounded_num**2
```

```
     print(f"sum_of_squares: {sum_of_squares}")
   print(f"return value: {sum_of_squares}")
   return sum_of_squares
```
**Code End**
Feedback: With the above function, we get sum_squares([1.4,4.2,0]) = 17.
   However, the test case is "sum_squares([1.4,4.2,0]) == 29". So the
   code doesn't pass the test case. Here's the log from the print()
   function in the code which can help debug:
**Log Start**
num: 1.4, rounded_num: 1
sum_of_squares: 1
num: 4.2, rounded_num: 4
sum_of_squares: 17
num: 0, rounded_num: 0
sum_of_squares: 17
return value: 17
**Log End**
The code above is wrong. Please find the bug by checking the test case
   and the log and fix it.
Explain both the test case and the log with the wrong result, compare the
    explanation of the log to the explanation of the test case to find
   the bug:
First, we explain the test case step by step. The test case is
   sum_squares([1.4,4.2,0]) == 29 because: We round 1.4 to the upper
   integer 2 and the square of 2 is 2**2 = 4, we round 4.2 to the upper
   integer 5 and the square of 5 is 5**2 = 25, we round 0 to the upper
   integer 0 and the square of 0 is 0**2 = 0. The sum of squares are 4 +
    25 + 0 = 29, which is the same to the ground truth 29 in the test
   case "sum_squares([1.4,4.2,0]) == 29".
Second, we explain the log line by line and the wrong result we got from
   the wrong code:
"num: 1.4, rounded_num: 1": The first num of input is 1.4, and round it
   to 1. From here, we find the bug. It's wrong because we should round
   1.4 to its upper integer 2 instead of 1.
"sum_of_squares: 1": This step is wrong because the right rounded num
   should be 2. The square is 2**2 = 4, so the sum of squares is 4.
"num: 4.2, rounded_num: 4": The second num of input is 4.2, we round it
   to 4. From here, we find the bug. It's wrong because we should round
   4.2 to its upper integer 5 instead of 4.
"sum_of_squares: 17": This step is wrong because the right rounded num
   should be 5. The square is 5**2 = 25, so the sum of squares is 4 + 25
    = 29.
"num: 0, rounded_num: 0": The third num is 0, and we round it to 0.
"sum_of_squares: 17": The rounded num is 0 and the square is 0**2 = 0.
   This step is wrong, so the sum of squares should be 4 + 25 + 0 = 29.
Finally, we compare the explanation of the test case with the log and the
    wrong result we got. We find the bug in the code: "rounded_num = int
   (num)" is wrong because from the log we found it doesn't round the
   number to its upper integer. We should round the input numbers to
   their upper integers. 1.4 should be rounded to 2 instead of 1, 4.2
   should be rounded to 5 instead of 4.
We should fix this bug according the above explanation:
**Code Start**
```
class Solution:
  def sum_squares(self, lst: List[int]) -> int:
    import math
    sum_of_squares = 0
    for num in lst:
      rounded_num = math.ceil(num)
```

```
        sum_of_squares += rounded_num**2
    return sum_of_squares
**Code End**
**Task End**

**Task Start**
**Description Start**
[Your Problem]
**Description End**
**Code Start**
[Generated Code]
**Code End**
Feedback: Don't change the code, but add some print() functions to print
    intermediate variables for debugging.
**Code Start**
[Code with added print statements]
**Code End**
[Feedback with test case and log]
**Log Start**
[Log]
**Log End**
[Explanation]
**Code Start**
[Corrected Code]
**Code End**
```

## B    LEETCODE DATASETS

Every problem collected in our dataset includes the following parts: id, URL (used to submit), released time, level (easy, medium, hard), title, description, examples (which includes input, expected output, explanation), constraints, header of the solution class in 12 languages, cues, pass rate in the parsing time, illustrations for easy understanding and one solution which could pass all test cases from the platform. Some of the parts are not used in out experiments but we will all release them for the future research.

## C    MORE EXAMPLES

We present an additional example in which the problem can be accessed through this URL. This particular problem requires the utilization of dynamic programming techniques for its solution. It takes print debugging 3 rounds to pass all the test cases but other methods all fail. The complete prompt[9] for all the rounds of this problem is as follows:

```
## ROUND 1
### PROMPT
**Task Start**
**Description Start**
You are given a list of numbers.
You need to return the sum of squared numbers in the given list, round
    each element in the list to the upper int(Ceiling) first.
class Solution:
    def sum_squares(self, lst: List[int]) -> int:
**Description End**
**Code Start**
class Solution:
    def sum_squares(self, lst: List[int]) -> int:
```

---

[9]We use "## ROUND X" to seperate each round, also "### PROMPT" and "### RAW COMPLETION" to represent the whole prompt sent to the model and the raw respond from the model on each round. They should be excluded from the prompt but we add them here only for clarity. The same for Appendix D.

```
        sum_of_squares = 0
        for num in lst:
            rounded_num = int(num)
            sum_of_squares += rounded_num**2
        return sum_of_squares
```
**Code End**
Feedback: Don't change the code, but add some print() functions to print
    intermediate variables for debugging.
**Code Start**
```
class Solution:
    def sum_squares(self, lst: List[int]) -> int:
        sum_of_squares = 0
        for num in lst:
            rounded_num = int(num)
            print(f"num: {num}, rounded_num: {rounded_num}")
            sum_of_squares += rounded_num**2
            print(f"sum_of_squares: {sum_of_squares}")
        print(f"return value: {sum_of_squares}")
        return sum_of_squares
```
**Code End**
Feedback: With the above function, we get sum_squares([1.4,4.2,0]) = 17.
    However, the test case is "sum_squares([1.4,4.2,0]) == 29". So the
    code doesn't pass the test case. Here's the log from the print()
    function in the code which can help debug:
**Log Start**
num: 1.4, rounded_num: 1
sum_of_squares: 1
num: 4.2, rounded_num: 4
sum_of_squares: 17
num: 0, rounded_num: 0
sum_of_squares: 17
return value: 17
**Log End**
The code above is wrong. Please find the bug by checking the test case
    and the log and fix it.
Explain both the test case and the log with the wrong result, compare the
     explanation of the log to the explanation of the test case to find
    the bug:
First, we explain the test case step by step. The test case is
    sum_squares([1.4,4.2,0]) == 29 because: We round 1.4 to the upper
    integer 2 and the square of 2 is 2**2 = 4, we round 4.2 to the upper
    integer 5 and the square of 5 is 5**2 = 25, we round 0 to the upper
    integer 0 and the square of 0 is 0**2 = 0. The sum of squares are 4 +
     25 + 0 = 29, which is the same to the ground truth 29 in the test
    case "sum_squares([1.4,4.2,0]) == 29".
Second, we explain the log line by line and the wrong result we got from
    the wrong code:
"num: 1.4, rounded_num: 1": The first num of input is 1.4, and round it
    to 1. From here, we find the bug. It's wrong because we should round
    1.4 to its upper integer 2 instead of 1.
"sum_of_squares: 1": This step is wrong because the right rounded num
    should be 2. The square is 2**2 = 4, so the sum of squares is 4.
"num: 4.2, rounded_num: 4": The second num of input is 4.2, we round it
    to 4. From here, we find the bug. It's wrong because we should round
    4.2 to its upper integer 5 instead of 4.
"sum_of_squares: 17": This step is wrong because the right rounded num
    should be 5. The square is 5**2 = 25, so the sum of squares is 4 + 25
     = 29.
"num: 0, rounded_num: 0": The third num is 0, and we round it to 0.

"sum_of_squares: 17": The rounded num is 0 and the square is 0**2 = 0.
  This step is wrong, so the sum of squares should be 4 + 25 + 0 = 29.
Finally, we compare the explanation of the test case with the log and the
  wrong result we got. We find the bug in the code: "rounded_num = int
  (num)" is wrong because from the log we found it doesn't round the
  number to its upper integer. We should round the input numbers to
  their upper integers. 1.4 should be rounded to 2 instead of 1, 4.2
  should be rounded to 5 instead of 4.
We should fix this bug according the above explanation:
**Code Start**

```
class Solution:
    def sum_squares(self, lst: List[int]) -> int:
        import math
        sum_of_squares = 0
        for num in lst:
            rounded_num = math.ceil(num)
            sum_of_squares += rounded_num**2
        return sum_of_squares
```

**Code End**
**Task End**

**Task Start**
**Description Start**
You are given a 0-indexed string s and a dictionary of words dictionary.
  You have to break s into one or more non-overlapping substrings such
  that each substring is present in dictionary. There may be some extra
   characters in s which are not present in any of the substrings.

Return the minimum number of extra characters left over if you break up s
  optimally.
**Description End**
**Code Start**

```
class Solution:
    def minExtraChar(self, s: str, dictionary: List[str]) -> int:
        n = len(s)
        dp = [0] * (n + 1)
        dp[0] = 0

        for i in range(1, n + 1):
            dp[i] = dp[i - 1] + 1
            for word in dictionary:
                if s[i - len(word):i] == word:
                    dp[i] = min(dp[i], dp[i - len(word)])
            print(f"dp[{i}]: {dp[i]}")

        print(f"return value: {dp[n] - 1}")
        return dp[n] - 1
```

**Code End**
Feedback: With the above function, we get minExtraChar("leetscode", ["
  leet","code","leetcode"]) = 0. However, the test case is "
  minExtraChar("leetscode", ["leet","code","leetcode"]) == 1". So the
  code doesn't pass the test case. Here's the log from the print()
  function in the code which can help debug:
**Log Start**
dp[1]: 1
dp[2]: 2
dp[3]: 3
dp[4]: 0
dp[5]: 1

```
dp[6]: 2
dp[7]: 3
dp[8]: 4
dp[9]: 1
return value: 0
```
**Log End**
The code above is wrong. Please find the bug by checking the test case
    and the log and fix it.
Explain both the test case and the log with the wrong answer, compare the
     explanation of the log to the explanation of the test case to find
    the bug:
### RAW COMPLETION
First, we explain the test case step by step. The test case is
    minExtraChar("leetscode", ["leet","code","leetcode"]) == 1 because:
    We can break the string "leetscode" into substrings "leet" and "code"
     which are present in the dictionary. There is one extra character 's
    ' left over, so the minimum number of extra characters left over is
    1, which is the same as the ground truth 1 in the test case "
    minExtraChar("leetscode", ["leet","code","leetcode"]) == 1".

Second, we explain the log line by line and the wrong result we got from
    the wrong code:
"dp[1]: 1", "dp[2]: 2", "dp[3]: 3": These steps are correct because there
     are no substrings in the dictionary that can be formed from the
    first 1, 2, or 3 characters of the string "leetscode".
"dp[4]: 0": This step is correct because the substring "leet" is present
    in the dictionary, so there are no extra characters left over for the
     first 4 characters of the string "leetscode".
"dp[5]: 1", "dp[6]: 2", "dp[7]: 3": These steps are correct because there
     are no substrings in the dictionary that can be formed from the
    characters between the substring "leet" and the substring "code" in
    the string "leetscode".
"dp[8]: 4": This step is wrong because the substring "code" is present in
     the dictionary, so there should be only 1 extra character left over
    (the 's' between "leet" and "code") for the first 8 characters of the
     string "leetscode".
"dp[9]: 1": This step is correct because there are no extra characters
    left over for the entire string "leetscode" after breaking it into
    substrings "leet" and "code".
"return value: 0": This step is wrong because the minimum number of extra
     characters left over should be 1, not 0.

Finally, we compare the explanation of the test case with the log and the
     wrong result we got. We find the bug in the code: The bug is in the
    line "dp[i] = min(dp[i], dp[i - len(word)])". The code is not
    considering the extra characters left over when breaking the string
    into substrings. We should add the number of extra characters left
    over to the dp[i] value when updating it.

We should fix this bug according to the above explanation:
**Code Start**
```
class Solution:
    def minExtraChar(self, s: str, dictionary: List[str]) -> int:
        n = len(s)
        dp = [0] * (n + 1)
        dp[0] = 0

        for i in range(1, n + 1):
            dp[i] = dp[i - 1] + 1
```

```
            for word in dictionary:
                if s[i - len(word):i] == word:
                    dp[i] = min(dp[i], dp[i - len(word)] + len(word) -
    len(s[i - len(word):i]))
            print(f"dp[{i}]: {dp[i]}")

        print(f"return value: {dp[n] - 1}")
        return dp[n] - 1
```
**Code End**
### RESULTS
[Feedback from Leetcode online judging system]
Input:
"leetscode"
["leet","code","leetcode"]
Output:
0
Expected answer:
1

## ROUND 2
### PROMPT
**Task Start**
**Description Start**
You are given a list of numbers.
You need to return the sum of squared numbers in the given list, round
    each element in the list to the upper int(Ceiling) first.
```
class Solution:
    def sum_squares(self, lst: List[int]) -> int:
```
**Description End**
**Code Start**
```
class Solution:
    def sum_squares(self, lst: List[int]) -> int:
        sum_of_squares = 0
        for num in lst:
            rounded_num = int(num)
            sum_of_squares += rounded_num**2
        return sum_of_squares
```
**Code End**
Feedback: Don't change the code, but add some print() functions to print
    intermediate variables for debugging.
**Code Start**
```
class Solution:
    def sum_squares(self, lst: List[int]) -> int:
        sum_of_squares = 0
        for num in lst:
            rounded_num = int(num)
            print(f"num: {num}, rounded_num: {rounded_num}")
            sum_of_squares += rounded_num**2
            print(f"sum_of_squares: {sum_of_squares}")
        print(f"return value: {sum_of_squares}")
        return sum_of_squares
```
**Code End**
Feedback: With the above function, we get sum_squares([1.4,4.2,0]) = 17.
    However, the test case is "sum_squares([1.4,4.2,0]) == 29". So the
    code doesn't pass the test case. Here's the log from the print()
    function in the code which can help debug:
**Log Start**
num: 1.4, rounded_num: 1
sum_of_squares: 1

num: 4.2, rounded_num: 4
sum_of_squares: 17
num: 0, rounded_num: 0
sum_of_squares: 17
return value: 17
**Log End**
The code above is wrong. Please find the bug by checking the test case
    and the log and fix it.
Explain both the test case and the log with the wrong result, compare the
     explanation of the log to the explanation of the test case to find
     the bug:
First, we explain the test case step by step. The test case is
    sum_squares([1.4,4.2,0]) == 29 because: We round 1.4 to the upper
    integer 2 and the square of 2 is 2**2 = 4, we round 4.2 to the upper
    integer 5 and the square of 5 is 5**2 = 25, we round 0 to the upper
    integer 0 and the square of 0 is 0**2 = 0. The sum of squares are 4 +
     25 + 0 = 29, which is the same to the ground truth 29 in the test
    case "sum_squares([1.4,4.2,0]) == 29".
Second, we explain the log line by line and the wrong result we got from
    the wrong code:
"num: 1.4, rounded_num: 1": The first num of input is 1.4, and round it
    to 1. From here, we find the bug. It's wrong because we should round
    1.4 to its upper integer 2 instead of 1.
"sum_of_squares: 1": This step is wrong because the right rounded num
    should be 2. The square is 2**2 = 4, so the sum of squares is 4.
"num: 4.2, rounded_num: 4": The second num of input is 4.2, we round it
    to 4. From here, we find the bug. It's wrong because we should round
    4.2 to its upper integer 5 instead of 4.
"sum_of_squares: 17": This step is wrong because the right rounded num
    should be 5. The square is 5**2 = 25, so the sum of squares is 4 + 25
     = 29.
"num: 0, rounded_num: 0": The third num is 0, and we round it to 0.
"sum_of_squares: 17": The rounded num is 0 and the square is 0**2 = 0.
    This step is wrong, so the sum of squares should be 4 + 25 + 0 = 29.
Finally, we compare the explanation of the test case with the log and the
     wrong result we got. We find the bug in the code: "rounded_num = int
    (num)" is wrong because from the log we found it doesn't round the
    number to its upper integer. We should round the input numbers to
    their upper integers. 1.4 should be rounded to 2 instead of 1, 4.2
    should be rounded to 5 instead of 4.
We should fix this bug according the above explanation:
**Code Start**
class Solution:
    def sum_squares(self, lst: List[int]) -> int:
        import math
        sum_of_squares = 0
        for num in lst:
            rounded_num = math.ceil(num)
            sum_of_squares += rounded_num**2
        return sum_of_squares
**Code End**
**Task End**

**Task Start**
**Description Start**
You are given a 0-indexed string s and a dictionary of words dictionary.
    You have to break s into one or more non-overlapping substrings such
    that each substring is present in dictionary. There may be some extra
     characters in s which are not present in any of the substrings.

Return the minimum number of extra characters left over if you break up s
    optimally.
**Description End**
**Code Start**
```python
class Solution:
    def minExtraChar(self, s: str, dictionary: List[str]) -> int:
        n = len(s)
        dp = [0] * (n + 1)
        dp[0] = 0

        for i in range(1, n + 1):
            dp[i] = dp[i - 1] + 1
            for word in dictionary:
                if s[i - len(word):i] == word:
                    dp[i] = min(dp[i], dp[i - len(word)] + len(word) -
    len(s[i - len(word):i]))
            print(f"dp[{i}]: {dp[i]}")

        print(f"return value: {dp[n] - 1}")
        return dp[n] - 1
```
**Code End**
Feedback: With the above function, we get minExtraChar("leetscode", ["
    leet","code","leetcode"]) = 0. However, the test case is "
    minExtraChar("leetscode", ["leet","code","leetcode"]) == 1". So the
    code doesn't pass the test case. Here's the log from the print()
    function in the code which can help debug:
**Log Start**
dp[1]: 1
dp[2]: 2
dp[3]: 3
dp[4]: 0
dp[5]: 1
dp[6]: 2
dp[7]: 3
dp[8]: 4
dp[9]: 1
return value: 0
**Log End**
The code above is wrong. Please find the bug by checking the test case
    and the log and fix it.
Explain both the test case and the log with the wrong answer, compare the
     explanation of the log to the explanation of the test case to find
    the bug:
### RAW COMPLETION
First, we explain the test case step by step. The test case is
    minExtraChar("leetscode", ["leet","code","leetcode"]) == 1 because:
    We can break the string "leetscode" into substrings "leet" and "code"
     which are present in the dictionary. There is one extra character 's
    ' left over, so the minimum number of extra characters left over is
    1, which is the same as the ground truth 1 in the test case "
    minExtraChar("leetscode", ["leet","code","leetcode"]) == 1".

Second, we explain the log line by line and the wrong result we got from
    the wrong code:
"dp[1]: 1", "dp[2]: 2", "dp[3]: 3": These are the initial values of dp
    array for the first three characters of the string "leetscode".
"dp[4]: 0": This step is correct because the substring "leet" is present
    in the dictionary and there are no extra characters left over.

"dp[5]: 1", "dp[6]: 2", "dp[7]: 3": These are the values of dp array for
    the next three characters of the string "leetscode".
"dp[8]: 4": This step is wrong because the substring "code" is present in
    the dictionary and there should be only one extra character 's' left
    over, not 4.
"dp[9]: 1": This step is wrong because the final value of dp array should
    be 2 (1 extra character 's' left over), not 1.
"return value: 0": This step is wrong because the return value should be
    1 (minimum number of extra characters left over), not 0.

Finally, we compare the explanation of the test case with the log and the
    wrong result we got. We find the bug in the code: "dp[i] = min(dp[i
    ], dp[i - len(word)] + len(word) - len(s[i - len(word):i]))" is wrong
     because it calculates the minimum number of extra characters left
    over incorrectly. We should update the dp array by subtracting the
    length of the word from the current dp value instead of adding it.

We should fix this bug according to the above explanation:
**Code Start**
```
class Solution:
    def minExtraChar(self, s: str, dictionary: List[str]) -> int:
        n = len(s)
        dp = [0] * (n + 1)
        dp[0] = 0

        for i in range(1, n + 1):
            dp[i] = dp[i - 1] + 1
            for word in dictionary:
                if s[i - len(word):i] == word:
                    dp[i] = min(dp[i], dp[i - len(word)] - len(word) +
    len(s[i - len(word):i]))

        return dp[n] - 1
```
**Code End**
### RESULTS
[Feedback from Leetcode online judging system]
Input:
"leetscode"
["leet","code","leetcode"]
Output:
0
Expected answer:
1

## ROUND 3
### PROMPT
**Task Start**
**Description Start**
You are given a list of numbers.
You need to return the sum of squared numbers in the given list, round
    each element in the list to the upper int(Ceiling) first.
```
class Solution:
    def sum_squares(self, lst: List[int]) -> int:
```
**Description End**
**Code Start**
```
class Solution:
    def sum_squares(self, lst: List[int]) -> int:
        sum_of_squares = 0
        for num in lst:
```

```
                rounded_num = int(num)
                sum_of_squares += rounded_num**2
            return sum_of_squares
```
**Code End**
Feedback: Don't change the code, but add some print() functions to print
    intermediate variables for debugging.
**Code Start**
```
class Solution:
    def sum_squares(self, lst: List[int]) -> int:
        sum_of_squares = 0
        for num in lst:
            rounded_num = int(num)
            print(f"num: {num}, rounded_num: {rounded_num}")
            sum_of_squares += rounded_num**2
            print(f"sum_of_squares: {sum_of_squares}")
        print(f"return value: {sum_of_squares}")
        return sum_of_squares
```
**Code End**
Feedback: With the above function, we get sum_squares([1.4,4.2,0]) = 17.
    However, the test case is "sum_squares([1.4,4.2,0]) == 29". So the
    code doesn't pass the test case. Here's the log from the print()
    function in the code which can help debug:
**Log Start**
num: 1.4, rounded_num: 1
sum_of_squares: 1
num: 4.2, rounded_num: 4
sum_of_squares: 17
num: 0, rounded_num: 0
sum_of_squares: 17
return value: 17
**Log End**
The code above is wrong. Please find the bug by checking the test case
    and the log and fix it.
Explain both the test case and the log with the wrong result, compare the
     explanation of the log to the explanation of the test case to find
    the bug:
First, we explain the test case step by step. The test case is
    sum_squares([1.4,4.2,0]) == 29 because: We round 1.4 to the upper
    integer 2 and the square of 2 is 2**2 = 4, we round 4.2 to the upper
    integer 5 and the square of 5 is 5**2 = 25, we round 0 to the upper
    integer 0 and the square of 0 is 0**2 = 0. The sum of squares are 4 +
     25 + 0 = 29, which is the same to the ground truth 29 in the test
    case "sum_squares([1.4,4.2,0]) == 29".
Second, we explain the log line by line and the wrong result we got from
    the wrong code:
"num: 1.4, rounded_num: 1": The first num of input is 1.4, and round it
    to 1. From here, we find the bug. It's wrong because we should round
    1.4 to its upper integer 2 instead of 1.
"sum_of_squares: 1": This step is wrong because the right rounded num
    should be 2. The square is 2**2 = 4, so the sum of squares is 4.
"num: 4.2, rounded_num: 4": The second num of input is 4.2, we round it
    to 4. From here, we find the bug. It's wrong because we should round
    4.2 to its upper integer 5 instead of 4.
"sum_of_squares: 17": This step is wrong because the right rounded num
    should be 5. The square is 5**2 = 25, so the sum of squares is 4 + 25
     = 29.
"num: 0, rounded_num: 0": The third num is 0, and we round it to 0.
"sum_of_squares: 17": The rounded num is 0 and the square is 0**2 = 0.
    This step is wrong, so the sum of squares should be 4 + 25 + 0 = 29.
```

Finally, we compare the explanation of the test case with the log and the
     wrong result we got. We find the bug in the code: "rounded_num = int
     (num)" is wrong because from the log we found it doesn't round the
     number to its upper integer. We should round the input numbers to
     their upper integers. 1.4 should be rounded to 2 instead of 1, 4.2
     should be rounded to 5 instead of 4.
We should fix this bug according the above explanation:
**Code Start**

```
class Solution:
    def sum_squares(self, lst: List[int]) -> int:
        import math
        sum_of_squares = 0
        for num in lst:
            rounded_num = math.ceil(num)
            sum_of_squares += rounded_num**2
        return sum_of_squares
```

**Code End**
**Task End**

**Task Start**
**Description Start**
You are given a 0-indexed string s and a dictionary of words dictionary.
    You have to break s into one or more non-overlapping substrings such
    that each substring is present in dictionary. There may be some extra
    characters in s which are not present in any of the substrings.

Return the minimum number of extra characters left over if you break up s
     optimally.
**Description End**
**Code Start**

```
class Solution:
    def minExtraChar(self, s: str, dictionary: List[str]) -> int:
        n = len(s)
        dp = [0] * (n + 1)
        dp[0] = 0

        for i in range(1, n + 1):
            dp[i] = dp[i - 1] + 1
            for word in dictionary:
                if s[i - len(word):i] == word:
                    dp[i] = min(dp[i], dp[i - len(word)] - len(word) +
    len(s[i - len(word):i]))
            print(f"dp[{i}]: {dp[i]}")

        print(f"return value: {dp[n] - 1}")
        return dp[n] - 1
```

**Code End**
Feedback: With the above function, we get minExtraChar("leetscode", ["
    leet","code","leetcode"]) = 0. However, the test case is "
    minExtraChar("leetscode", ["leet","code","leetcode"]) == 1". So the
    code doesn't pass the test case. Here's the log from the print()
    function in the code which can help debug:
**Log Start**
dp[1]: 1
dp[2]: 2
dp[3]: 3
dp[4]: 0
dp[5]: 1
dp[6]: 2

```
dp[7]: 3
dp[8]: 4
dp[9]: 1
return value: 0
**Log End**
```

The code above is wrong. Please find the bug by checking the test case
    and the log and fix it.
Explain both the test case and the log with the wrong answer, compare the
    explanation of the log to the explanation of the test case to find
    the bug:
### RAW COMPLETION
First, we explain the test case step by step. The test case is
    minExtraChar("leetscode", ["leet","code","leetcode"]) == 1 because:
    We can break the string "leetscode" into substrings "leet" and "code"
     which are present in the dictionary. There is one extra character 's
    ' left over, so the minimum number of extra characters left over is
    1, which is the same as the ground truth 1 in the test case "
    minExtraChar("leetscode", ["leet","code","leetcode"]) == 1".

Second, we explain the log line by line and the wrong result we got from
    the wrong code:
"dp[1]: 1", "dp[2]: 2", "dp[3]: 3": These are the initial values of dp
    for the first three characters of the string "leetscode".
"dp[4]: 0": This step is correct because the substring "leet" is present
    in the dictionary, so there are no extra characters left over at this
     point.
"dp[5]: 1", "dp[6]: 2", "dp[7]: 3": These steps are wrong because the
    substring "code" is present in the dictionary, so there should be
    only one extra character left over (the 's' between "leet" and "code
    ").
"dp[8]: 4": This step is wrong because the substring "code" is present in
     the dictionary, so there should be only one extra character left
    over (the 's' between "leet" and "code").
"dp[9]: 1": This step is correct because the substring "code" is present
    in the dictionary, so there is only one extra character left over (
    the 's' between "leet" and "code").
"return value: 0": This step is wrong because the correct return value
    should be 1, as there is one extra character left over (the 's'
    between "leet" and "code").

Finally, we compare the explanation of the test case with the log and the
     wrong result we got. We find the bug in the code: "dp[i] = min(dp[i
    ], dp[i - len(word)] - len(word) + len(s[i - len(word):i]))" is wrong
     because it doesn't correctly calculate the minimum number of extra
    characters left over. We should fix this bug according to the above
    explanation:

**Code Start**
```
class Solution:
    def minExtraChar(self, s: str, dictionary: List[str]) -> int:
        n = len(s)
        dp = [0] * (n + 1)
        dp[0] = 0

        for i in range(1, n + 1):
            dp[i] = dp[i - 1] + 1
            for word in dictionary:
                if s[i - len(word):i] == word:
                    dp[i] = min(dp[i], dp[i - len(word)])
```

```
        print(f"dp[{i}]: {dp[i]}")

      print(f"return value: {dp[n]}")
      return dp[n]
```
**Code End**
**Task End**
### RESULTS
[Feedback from Leetcode online judging system]
succeed

# D  ALL ROUNDS

## ROUND 1
### PROMPT
**Task Start**
**Description Start**
You are given a list of numbers.
You need to return the sum of squared numbers in the given list, round
    each element in the list to the upper int(Ceiling) first.
class Solution:
    def sum_squares(self, lst: List[int]) -> int:
**Description End**
**Code Start**
```
class Solution:
    def sum_squares(self, lst: List[int]) -> int:
        sum_of_squares = 0
        for num in lst:
            rounded_num = int(num)
            sum_of_squares += rounded_num**2
        return sum_of_squares
```
**Code End**
Feedback: Don't change the code, but add some print() functions to print
    intermediate variables for debugging.
**Code Start**
```
class Solution:
    def sum_squares(self, lst: List[int]) -> int:
        sum_of_squares = 0
        for num in lst:
            rounded_num = int(num)
            print(f"num: {num}, rounded_num: {rounded_num}")
            sum_of_squares += rounded_num**2
            print(f"sum_of_squares: {sum_of_squares}")
        print(f"return value: {sum_of_squares}")
        return sum_of_squares
```
**Code End**
Feedback: With the above function, we get sum_squares([1.4,4.2,0]) = 17.
    However, the test case is "sum_squares([1.4,4.2,0]) == 29". So the
    code doesn't pass the test case. Here's the log from the print()
    function in the code which can help debug:
**Log Start**
num: 1.4, rounded_num: 1
sum_of_squares: 1
num: 4.2, rounded_num: 4
sum_of_squares: 17
num: 0, rounded_num: 0
sum_of_squares: 17
return value: 17
**Log End**

The code above is wrong. Please find the bug by checking the test case
    and the log and fix it.
Explain both the test case and the log with the wrong result, compare the
     explanation of the log to the explanation of the test case to find
     the bug:
First, we explain the test case step by step. The test case is
    sum_squares([1.4,4.2,0]) == 29 because: We round 1.4 to the upper
    integer 2 and the square of 2 is 2**2 = 4, we round 4.2 to the upper
    integer 5 and the square of 5 is 5**2 = 25, we round 0 to the upper
    integer 0 and the square of 0 is 0**2 = 0. The sum of squares are 4 +
     25 + 0 = 29, which is the same to the ground truth 29 in the test
    case "sum_squares([1.4,4.2,0]) == 29".
Second, we explain the log line by line and the wrong result we got from
    the wrong code:
"num: 1.4, rounded_num: 1": The first num of input is 1.4, and round it
    to 1. From here, we find the bug. It's wrong because we should round
    1.4 to its upper integer 2 instead of 1.
"sum_of_squares: 1": This step is wrong because the right rounded num
    should be 2. The square is 2**2 = 4, so the sum of squares is 4.
"num: 4.2, rounded_num: 4": The second num of input is 4.2, we round it
    to 4. From here, we find the bug. It's wrong because we should round
    4.2 to its upper integer 5 instead of 4.
"sum_of_squares: 17": This step is wrong because the right rounded num
    should be 5. The square is 5**2 = 25, so the sum of squares is 4 + 25
     = 29.
"num: 0, rounded_num: 0": The third num is 0, and we round it to 0.
"sum_of_squares: 17": The rounded num is 0 and the square is 0**2 = 0.
    This step is wrong, so the sum of squares should be 4 + 25 + 0 = 29.
Finally, we compare the explanation of the test case with the log and the
     wrong result we got. We find the bug in the code: "rounded_num = int
    (num)" is wrong because from the log we found it doesn't round the
    number to its upper integer. We should round the input numbers to
    their upper integers. 1.4 should be rounded to 2 instead of 1, 4.2
    should be rounded to 5 instead of 4.
We should fix this bug according the above explanation:
**Code Start**
```
class Solution:
    def sum_squares(self, lst: List[int]) -> int:
        import math
        sum_of_squares = 0
        for num in lst:
            rounded_num = math.ceil(num)
            sum_of_squares += rounded_num**2
        return sum_of_squares
```
**Code End**
**Task End**

**Task Start**
**Description Start**
You are given two 0-indexed integer permutations A and B of length n.

A prefix common array of A and B is an array C such that C[i] is equal to
     the count of numbers that are present at or before the index i in
    both A and B.

Return the prefix common array of A and B.

A sequence of n integers is called apermutation if it contains all
    integers from 1 to n exactly once.

**Description End**
**Code Start**
```
class Solution:
    def findThePrefixCommonArray(self, A: List[int], B: List[int]) ->
    List[int]:
        common_array = []
        common_count = 0
        common_elements = set()

        for i in range(len(A)):
            if A[i] in B[:i+1]:
                common_elements.add(A[i])
            common_count = len(common_elements)
            common_array.append(common_count)
            print(f"i: {i}, A[i]: {A[i]}, B[:i+1]: {B[:i+1]},
    common_elements: {common_elements}, common_count: {common_count},
    common_array: {common_array}")

        print(f"return value: {common_array}")
        return common_array
```
**Code End**
Feedback: With the above function, we get findThePrefixCommonArray
    ([1,3,2,4], [3,1,2,4]) = [0, 1, 2, 3]. However, the test case is "
    findThePrefixCommonArray([1,3,2,4], [3,1,2,4]) == [0,2,3,4]". So the
    code doesn't pass the test case. Here's the log from the print()
    function in the code which can help debug:
**Log Start**
i: 0, A[i]: 1, B[:i+1]: [3], common_elements: set(), common_count: 0,
    common_array: [0]
i: 1, A[i]: 3, B[:i+1]: [3, 1], common_elements: {3}, common_count: 1,
    common_array: [0, 1]
i: 2, A[i]: 2, B[:i+1]: [3, 1, 2], common_elements: {2, 3}, common_count:
     2, common_array: [0, 1, 2]
i: 3, A[i]: 4, B[:i+1]: [3, 1, 2, 4], common_elements: {2, 3, 4},
    common_count: 3, common_array: [0, 1, 2, 3]
return value: [0, 1, 2, 3]
**Log End**
The code above is wrong. Please find the bug by checking the test case
    and the log and fix it.
Explain both the test case and the log with the wrong answer, compare the
     explanation of the log to the explanation of the test case to find
    the bug:
### RAW COMPLETION
First, we explain the test case step by step. The test case is
    findThePrefixCommonArray([1,3,2,4], [3,1,2,4]) == [0,2,3,4] because:
At index 0, there are no common elements in A and B, so the common array
    value is 0.
At index 1, there are 2 common elements (1 and 3) in A[:2] and B[:2], so
    the common array value is 2.
At index 2, there are 3 common elements (1, 2, and 3) in A[:3] and B[:3],
     so the common array value is 3.
At index 3, there are 4 common elements (1, 2, 3, and 4) in A[:4] and B
    [:4], so the common array value is 4.
The prefix common array is [0, 2, 3, 4], which is the same as the ground
    truth in the test case "findThePrefixCommonArray([1,3,2,4],
    [3,1,2,4]) == [0,2,3,4]".

Second, we explain the log line by line and the wrong result we got from
    the wrong code:

"i: 0, A[i]: 1, B[:i+1]: [3], common_elements: set(), common_count: 0,
    common_array: [0]": At index 0, there are no common elements in A and
     B, so the common array value is 0. This step is correct.
"i: 1, A[i]: 3, B[:i+1]: [3, 1], common_elements: {3}, common_count: 1,
    common_array: [0, 1]": At index 1, there are 2 common elements (1 and
     3) in A[:2] and B[:2], but the code only finds 1 common element (3).
     This step is wrong.
"i: 2, A[i]: 2, B[:i+1]: [3, 1, 2], common_elements: {2, 3}, common_count
    : 2, common_array: [0, 1, 2]": At index 2, there are 3 common
    elements (1, 2, and 3) in A[:3] and B[:3], but the code only finds 2
    common elements (2 and 3). This step is wrong.
"i: 3, A[i]: 4, B[:i+1]: [3, 1, 2, 4], common_elements: {2, 3, 4},
    common_count: 3, common_array: [0, 1, 2, 3]": At index 3, there are 4
     common elements (1, 2, 3, and 4) in A[:4] and B[:4], but the code
    only finds 3 common elements (2, 3, and 4). This step is wrong.
"return value: [0, 1, 2, 3]": The code returns the wrong prefix common
    array [0, 1, 2, 3] instead of the correct one [0, 2, 3, 4].

Finally, we compare the explanation of the test case with the log and the
     wrong result we got. We find the bug in the code: "if A[i] in B[:i
    +1]:" is wrong because from the log we found it doesn't find all the
    common elements at or before the index i in both A and B. We should
    change the code to find the common elements correctly.

We should fix this bug according to the above explanation:
**Code Start**
```
class Solution:
    def findThePrefixCommonArray(self, A: List[int], B: List[int]) ->
    List[int]:
        common_array = []
        common_count = 0
        common_elements = set()

        for i in range(len(A)):
            if B[i] in A[:i+1]:
                common_elements.add(B[i])
            common_count = len(common_elements)
            common_array.append(common_count)

        return common_array
```
**Code End**
**Task End**
### RESULTS
[Feedback from Leetcode online judging system]
wrong answer
Input:
[1,3,2,4]
[3,1,2,4]
Output:
[0,1,2,3]
Expected answer:
[0,2,3,4]

## ROUND 2
### PROMPT
**Task Start**
**Description Start**
You are given a list of numbers.

You need to return the sum of squared numbers in the given list, round
    each element in the list to the upper int(Ceiling) first.
class Solution:
    def sum_squares(self, lst: List[int]) -> int:
**Description End**
**Code Start**
class Solution:
    def sum_squares(self, lst: List[int]) -> int:
        sum_of_squares = 0
        for num in lst:
            rounded_num = int(num)
            sum_of_squares += rounded_num**2
        return sum_of_squares
**Code End**
Feedback: Don't change the code, but add some print() functions to print
    intermediate variables for debugging.
**Code Start**
class Solution:
    def sum_squares(self, lst: List[int]) -> int:
        sum_of_squares = 0
        for num in lst:
            rounded_num = int(num)
            print(f"num: {num}, rounded_num: {rounded_num}")
            sum_of_squares += rounded_num**2
            print(f"sum_of_squares: {sum_of_squares}")
        print(f"return value: {sum_of_squares}")
        return sum_of_squares
**Code End**
Feedback: With the above function, we get sum_squares([1.4,4.2,0]) = 17.
    However, the test case is "sum_squares([1.4,4.2,0]) == 29". So the
    code doesn't pass the test case. Here's the log from the print()
    function in the code which can help debug:
**Log Start**
num: 1.4, rounded_num: 1
sum_of_squares: 1
num: 4.2, rounded_num: 4
sum_of_squares: 17
num: 0, rounded_num: 0
sum_of_squares: 17
return value: 17
**Log End**
The code above is wrong. Please find the bug by checking the test case
    and the log and fix it.
Explain both the test case and the log with the wrong result, compare the
     explanation of the log to the explanation of the test case to find
    the bug:
First, we explain the test case step by step. The test case is
    sum_squares([1.4,4.2,0]) == 29 because: We round 1.4 to the upper
    integer 2 and the square of 2 is 2**2 = 4, we round 4.2 to the upper
    integer 5 and the square of 5 is 5**2 = 25, we round 0 to the upper
    integer 0 and the square of 0 is 0**2 = 0. The sum of squares are 4 +
     25 + 0 = 29, which is the same to the ground truth 29 in the test
    case "sum_squares([1.4,4.2,0]) == 29".
Second, we explain the log line by line and the wrong result we got from
    the wrong code:
"num: 1.4, rounded_num: 1": The first num of input is 1.4, and round it
    to 1. From here, we find the bug. It's wrong because we should round
    1.4 to its upper integer 2 instead of 1.

"sum_of_squares: 1": This step is wrong because the right rounded num
    should be 2. The square is 2**2 = 4, so the sum of squares is 4.
"num: 4.2, rounded_num: 4": The second num of input is 4.2, we round it
    to 4. From here, we find the bug. It's wrong because we should round
    4.2 to its upper integer 5 instead of 4.
"sum_of_squares: 17": This step is wrong because the right rounded num
    should be 5. The square is 5**2 = 25, so the sum of squares is 4 + 25
     = 29.
"num: 0, rounded_num: 0": The third num is 0, and we round it to 0.
"sum_of_squares: 17": The rounded num is 0 and the square is 0**2 = 0.
    This step is wrong, so the sum of squares should be 4 + 25 + 0 = 29.
Finally, we compare the explanation of the test case with the log and the
     wrong result we got. We find the bug in the code: "rounded_num = int
    (num)" is wrong because from the log we found it doesn't round the
    number to its upper integer. We should round the input numbers to
    their upper integers. 1.4 should be rounded to 2 instead of 1, 4.2
    should be rounded to 5 instead of 4.
We should fix this bug according the above explanation:
**Code Start**
class Solution:
    def sum_squares(self, lst: List[int]) -> int:
        import math
        sum_of_squares = 0
        for num in lst:
            rounded_num = math.ceil(num)
            sum_of_squares += rounded_num**2
        return sum_of_squares
**Code End**
**Task End**

**Task Start**
**Description Start**
You are given two 0-indexed integer permutations A and B of length n.

A prefix common array of A and B is an array C such that C[i] is equal to
     the count of numbers that are present at or before the index i in
    both A and B.

Return the prefix common array of A and B.

A sequence of n integers is called apermutation if it contains all
    integers from 1 to n exactly once.
**Description End**
**Code Start**
class Solution:
    def findThePrefixCommonArray(self, A: List[int], B: List[int]) ->
    List[int]:
        common_array = []
        common_count = 0
        common_elements = set()

        for i in range(len(A)):
            if B[i] in A[:i+1]:
                common_elements.add(B[i])
            common_count = len(common_elements)
            common_array.append(common_count)
            print(f"i: {i}, A[:i+1]: {A[:i+1]}, B[i]: {B[i]},
    common_elements: {common_elements}, common_count: {common_count},
    common_array: {common_array}")

```
        print(f"return value: {common_array}")
        return common_array
```
**Code End**

Feedback: With the above function, we get findThePrefixCommonArray
([1,3,2,4], [3,1,2,4]) = [0, 1, 2, 3]. However, the test case is "
findThePrefixCommonArray([1,3,2,4], [3,1,2,4]) == [0,2,3,4]". So the
code doesn't pass the test case. Here's the log from the print()
function in the code which can help debug:

**Log Start**

i: 0, A[:i+1]: [1], B[i]: 3, common_elements: set(), common_count: 0,
    common_array: [0]

i: 1, A[:i+1]: [1, 3], B[i]: 1, common_elements: {1}, common_count: 1,
    common_array: [0, 1]

i: 2, A[:i+1]: [1, 3, 2], B[i]: 2, common_elements: {1, 2}, common_count:
     2, common_array: [0, 1, 2]

i: 3, A[:i+1]: [1, 3, 2, 4], B[i]: 4, common_elements: {1, 2, 4},
    common_count: 3, common_array: [0, 1, 2, 3]

return value: [0, 1, 2, 3]

**Log End**

The code above is wrong. Please find the bug by checking the test case
    and the log and fix it.

Explain both the test case and the log with the wrong answer, compare the
     explanation of the log to the explanation of the test case to find
     the bug:

### RAW COMPLETION

First, we explain the test case step by step. The test case is
    findThePrefixCommonArray([1,3,2,4], [3,1,2,4]) == [0,2,3,4] because:
- At index 0, there are no common elements between A[:1] = [1] and B[:1]
    = [3], so the common array should be [0].
- At index 1, there are 2 common elements between A[:2] = [1, 3] and B
    [:2] = [3, 1], so the common array should be [0, 2].
- At index 2, there are 3 common elements between A[:3] = [1, 3, 2] and B
    [:3] = [3, 1, 2], so the common array should be [0, 2, 3].
- At index 3, there are 4 common elements between A[:4] = [1, 3, 2, 4]
    and B[:4] = [3, 1, 2, 4], so the common array should be [0, 2, 3, 4].

Second, we explain the log line by line and the wrong result we got from
    the wrong code:
- "i: 0, A[:i+1]: [1], B[i]: 3, common_elements: set(), common_count: 0,
    common_array: [0]": The first index is 0, A[:1] = [1], B[0] = 3,
    there are no common elements, so the common array is [0].
- "i: 1, A[:i+1]: [1, 3], B[i]: 1, common_elements: {1}, common_count: 1,
     common_array: [0, 1]": The second index is 1, A[:2] = [1, 3], B[1] =
     1, there is 1 common element (1), so the common array is [0, 1].
    From here, we find the bug. It's wrong because there should be 2
    common elements (1 and 3) instead of 1.
- "i: 2, A[:i+1]: [1, 3, 2], B[i]: 2, common_elements: {1, 2},
    common_count: 2, common_array: [0, 1, 2]": The third index is 2, A
    [:3] = [1, 3, 2], B[2] = 2, there are 2 common elements (1 and 2), so
     the common array is [0, 1, 2]. This step is correct.
- "i: 3, A[:i+1]: [1, 3, 2, 4], B[i]: 4, common_elements: {1, 2, 4},
    common_count: 3, common_array: [0, 1, 2, 3]": The fourth index is 3,
    A[:4] = [1, 3, 2, 4], B[3] = 4, there are 3 common elements (1, 2,
    and 4), so the common array is [0, 1, 2, 3]. This step is wrong
    because there should be 4 common elements (1, 2, 3, and 4) instead of
     3.

Finally, we compare the explanation of the test case with the log and the
    wrong result we got. We find the bug in the code: "if B[i] in A[:i
    +1]:" is wrong because from the log we found it doesn't count the
    common elements correctly. We should change the logic to count the
    common elements in A[:i+1] and B[:i+1] instead of checking if B[i] is
    in A[:i+1].

We should fix this bug according to the above explanation:

**Code Start**
```
class Solution:
    def findThePrefixCommonArray(self, A: List[int], B: List[int]) ->
    List[int]:
        common_array = []
        common_count = 0
        common_elements = set()

        for i in range(len(A)):
            if A[i] in B[:i+1]:
                common_elements.add(A[i])
            common_count = len(common_elements)
            common_array.append(common_count)

        return common_array
```
**Code End**
**Task End**
### RESULTS
[Feedback from Leetcode online judging system]
wrong answer
Input:
[1,3,2,4]
[3,1,2,4]
Output:
[0,1,2,3]
Expected answer:
[0,2,3,4]

## ROUND 3
### PROMPT
**Task Start**
**Description Start**
You are given a list of numbers.
You need to return the sum of squared numbers in the given list, round
    each element in the list to the upper int(Ceiling) first.
```
class Solution:
    def sum_squares(self, lst: List[int]) -> int:
```
**Description End**
**Code Start**
```
class Solution:
    def sum_squares(self, lst: List[int]) -> int:
        sum_of_squares = 0
        for num in lst:
            rounded_num = int(num)
            sum_of_squares += rounded_num**2
        return sum_of_squares
```
**Code End**
Feedback: Don't change the code, but add some print() functions to print
    intermediate variables for debugging.
**Code Start**

```
class Solution:
    def sum_squares(self, lst: List[int]) -> int:
        sum_of_squares = 0
        for num in lst:
            rounded_num = int(num)
            print(f"num: {num}, rounded_num: {rounded_num}")
            sum_of_squares += rounded_num**2
            print(f"sum_of_squares: {sum_of_squares}")
        print(f"return value: {sum_of_squares}")
        return sum_of_squares
```
**Code End**
Feedback: With the above function, we get sum_squares([1.4,4.2,0]) = 17.
    However, the test case is "sum_squares([1.4,4.2,0]) == 29". So the
    code doesn't pass the test case. Here's the log from the print()
    function in the code which can help debug:
**Log Start**
num: 1.4, rounded_num: 1
sum_of_squares: 1
num: 4.2, rounded_num: 4
sum_of_squares: 17
num: 0, rounded_num: 0
sum_of_squares: 17
return value: 17
**Log End**
The code above is wrong. Please find the bug by checking the test case
    and the log and fix it.
Explain both the test case and the log with the wrong result, compare the
     explanation of the log to the explanation of the test case to find
    the bug:
First, we explain the test case step by step. The test case is
    sum_squares([1.4,4.2,0]) == 29 because: We round 1.4 to the upper
    integer 2 and the square of 2 is 2**2 = 4, we round 4.2 to the upper
    integer 5 and the square of 5 is 5**2 = 25, we round 0 to the upper
    integer 0 and the square of 0 is 0**2 = 0. The sum of squares are 4 +
     25 + 0 = 29, which is the same to the ground truth 29 in the test
    case "sum_squares([1.4,4.2,0]) == 29".
Second, we explain the log line by line and the wrong result we got from
    the wrong code:
"num: 1.4, rounded_num: 1": The first num of input is 1.4, and round it
    to 1. From here, we find the bug. It's wrong because we should round
    1.4 to its upper integer 2 instead of 1.
"sum_of_squares: 1": This step is wrong because the right rounded num
    should be 2. The square is 2**2 = 4, so the sum of squares is 4.
"num: 4.2, rounded_num: 4": The second num of input is 4.2, we round it
    to 4. From here, we find the bug. It's wrong because we should round
    4.2 to its upper integer 5 instead of 4.
"sum_of_squares: 17": This step is wrong because the right rounded num
    should be 5. The square is 5**2 = 25, so the sum of squares is 4 + 25
     = 29.
"num: 0, rounded_num: 0": The third num is 0, and we round it to 0.
"sum_of_squares: 17": The rounded num is 0 and the square is 0**2 = 0.
    This step is wrong, so the sum of squares should be 4 + 25 + 0 = 29.
Finally, we compare the explanation of the test case with the log and the
     wrong result we got. We find the bug in the code: "rounded_num = int
    (num)" is wrong because from the log we found it doesn't round the
    number to its upper integer. We should round the input numbers to
    their upper integers. 1.4 should be rounded to 2 instead of 1, 4.2
    should be rounded to 5 instead of 4.
We should fix this bug according the above explanation:

```
**Code Start**
class Solution:
    def sum_squares(self, lst: List[int]) -> int:
        import math
        sum_of_squares = 0
        for num in lst:
            rounded_num = math.ceil(num)
            sum_of_squares += rounded_num**2
        return sum_of_squares
**Code End**
**Task End**

**Task Start**
**Description Start**
You are given two 0-indexed integer permutations A and B of length n.

A prefix common array of A and B is an array C such that C[i] is equal to
    the count of numbers that are present at or before the index i in
    both A and B.

Return the prefix common array of A and B.

A sequence of n integers is called apermutation if it contains all
    integers from 1 to n exactly once.
**Description End**
**Code Start**
class Solution:
    def findThePrefixCommonArray(self, A: List[int], B: List[int]) ->
    List[int]:
        common_array = []
        common_count = 0
        common_elements = set()

        for i in range(len(A)):
            if A[i] in B[:i+1]:
                common_elements.add(A[i])
            common_count = len(common_elements)
            common_array.append(common_count)
            print(f"i: {i}, A[i]: {A[i]}, B[:i+1]: {B[:i+1]},
    common_elements: {common_elements}, common_count: {common_count},
    common_array: {common_array}")

        print(f"return value: {common_array}")
        return common_array
**Code End**
```

Feedback: With the above function, we get findThePrefixCommonArray
    ([1,3,2,4], [3,1,2,4]) = [0, 1, 2, 3]. However, the test case is "
    findThePrefixCommonArray([1,3,2,4], [3,1,2,4]) == [0,2,3,4]". So the
    code doesn't pass the test case. Here's the log from the print()
    function in the code which can help debug:

```
**Log Start**
i: 0, A[i]: 1, B[:i+1]: [3], common_elements: set(), common_count: 0,
    common_array: [0]
i: 1, A[i]: 3, B[:i+1]: [3, 1], common_elements: {3}, common_count: 1,
    common_array: [0, 1]
i: 2, A[i]: 2, B[:i+1]: [3, 1, 2], common_elements: {2, 3}, common_count:
     2, common_array: [0, 1, 2]
i: 3, A[i]: 4, B[:i+1]: [3, 1, 2, 4], common_elements: {2, 3, 4},
    common_count: 3, common_array: [0, 1, 2, 3]
```

return value: [0, 1, 2, 3]
**Log End**
The code above is wrong. Please find the bug by checking the test case
    and the log and fix it.
Explain both the test case and the log with the wrong answer, compare the
     explanation of the log to the explanation of the test case to find
    the bug:
### RAW COMPLETION
First, we explain the test case step by step. The test case is
    findThePrefixCommonArray([1,3,2,4], [3,1,2,4]) == [0,2,3,4] because:
At index 0, there are no common elements in A and B, so the common array
    value is 0.
At index 1, there are 2 common elements (1 and 3) in A[:2] and B[:2], so
    the common array value is 2.
At index 2, there are 3 common elements (1, 2, and 3) in A[:3] and B[:3],
     so the common array value is 3.
At index 3, there are 4 common elements (1, 2, 3, and 4) in A[:4] and B
    [:4], so the common array value is 4.
The prefix common array is [0, 2, 3, 4], which is the same as the ground
    truth in the test case "findThePrefixCommonArray([1,3,2,4],
    [3,1,2,4]) == [0,2,3,4]".

Second, we explain the log line by line and the wrong result we got from
    the wrong code:
"i: 0, A[i]: 1, B[:i+1]: [3], common_elements: set(), common_count: 0,
    common_array: [0]": The first index is 0, A[0] is 1, and B[:1] is
    [3]. There are no common elements, so the common count is 0, and the
    common array is [0].
"i: 1, A[i]: 3, B[:i+1]: [3, 1], common_elements: {3}, common_count: 1,
    common_array: [0, 1]": The second index is 1, A[1] is 3, and B[:2] is
     [3, 1]. There is 1 common element (3), so the common count is 1, and
     the common array is [0, 1]. From here, we find the bug. It's wrong
    because there should be 2 common elements (1 and 3) instead of just
    1.
"i: 2, A[i]: 2, B[:i+1]: [3, 1, 2], common_elements: {2, 3}, common_count
    : 2, common_array: [0, 1, 2]": The third index is 2, A[2] is 2, and B
    [:3] is [3, 1, 2]. There are 2 common elements (2 and 3), so the
    common count is 2, and the common array is [0, 1, 2]. This step is
    wrong because there should be 3 common elements (1, 2, and 3) instead
     of just 2.
"i: 3, A[i]: 4, B[:i+1]: [3, 1, 2, 4], common_elements: {2, 3, 4},
    common_count: 3, common_array: [0, 1, 2, 3]": The fourth index is 3,
    A[3] is 4, and B[:4] is [3, 1, 2, 4]. There are 3 common elements (2,
     3, and 4), so the common count is 3, and the common array is [0, 1,
    2, 3]. This step is wrong because there should be 4 common elements
    (1, 2, 3, and 4) instead of just 3.

Finally, we compare the explanation of the test case with the log and the
     wrong result we got. We find the bug in the code: "if A[i] in B[:i
    +1]:" is wrong because from the log we found it doesn't count the
    common elements correctly. We should check if A[i] is in B and if B[i
    ] is in A to count the common elements correctly.

We should fix this bug according to the above explanation:

**Code Start**
class Solution:
    def findThePrefixCommonArray(self, A: List[int], B: List[int]) ->
    List[int]:

```
        common_array = []
        common_count = 0
        common_elements = set()

        for i in range(len(A)):
            if A[i] in B and B[i] in A:
                common_elements.add(A[i])
                common_elements.add(B[i])
            common_count = len(common_elements)
            common_array.append(common_count)

        return common_array
```
**Code End**
**Task End**
### RESULTS
[Feedback from Leetcode online judging system]
wrong answer
Input:
[1,3,2,4]
[3,1,2,4]
Output:
[2,2,3,4]
Expected answer:
[0,2,3,4]

## ROUND 4
### PROMPT
**Task Start**
**Description Start**
You are given a list of numbers.
You need to return the sum of squared numbers in the given list, round
    each element in the list to the upper int(Ceiling) first.
class Solution:
    def sum_squares(self, lst: List[int]) -> int:
**Description End**
**Code Start**
```
class Solution:
    def sum_squares(self, lst: List[int]) -> int:
        sum_of_squares = 0
        for num in lst:
            rounded_num = int(num)
            sum_of_squares += rounded_num**2
        return sum_of_squares
```
**Code End**
Feedback: Don't change the code, but add some print() functions to print
    intermediate variables for debugging.
**Code Start**
```
class Solution:
    def sum_squares(self, lst: List[int]) -> int:
        sum_of_squares = 0
        for num in lst:
            rounded_num = int(num)
            print(f"num: {num}, rounded_num: {rounded_num}")
            sum_of_squares += rounded_num**2
            print(f"sum_of_squares: {sum_of_squares}")
        print(f"return value: {sum_of_squares}")
        return sum_of_squares
```
**Code End**

Feedback: With the above function, we get sum_squares([1.4,4.2,0]) = 17.
    However, the test case is "sum_squares([1.4,4.2,0]) == 29". So the
    code doesn't pass the test case. Here's the log from the print()
    function in the code which can help debug:
**Log Start**
num: 1.4, rounded_num: 1
sum_of_squares: 1
num: 4.2, rounded_num: 4
sum_of_squares: 17
num: 0, rounded_num: 0
sum_of_squares: 17
return value: 17
**Log End**
The code above is wrong. Please find the bug by checking the test case
    and the log and fix it.
Explain both the test case and the log with the wrong result, compare the
     explanation of the log to the explanation of the test case to find
    the bug:
First, we explain the test case step by step. The test case is
    sum_squares([1.4,4.2,0]) == 29 because: We round 1.4 to the upper
    integer 2 and the square of 2 is 2**2 = 4, we round 4.2 to the upper
    integer 5 and the square of 5 is 5**2 = 25, we round 0 to the upper
    integer 0 and the square of 0 is 0**2 = 0. The sum of squares are 4 +
     25 + 0 = 29, which is the same to the ground truth 29 in the test
    case "sum_squares([1.4,4.2,0]) == 29".
Second, we explain the log line by line and the wrong result we got from
    the wrong code:
"num: 1.4, rounded_num: 1": The first num of input is 1.4, and round it
    to 1. From here, we find the bug. It's wrong because we should round
    1.4 to its upper integer 2 instead of 1.
"sum_of_squares: 1": This step is wrong because the right rounded num
    should be 2. The square is 2**2 = 4, so the sum of squares is 4.
"num: 4.2, rounded_num: 4": The second num of input is 4.2, we round it
    to 4. From here, we find the bug. It's wrong because we should round
    4.2 to its upper integer 5 instead of 4.
"sum_of_squares: 17": This step is wrong because the right rounded num
    should be 5. The square is 5**2 = 25, so the sum of squares is 4 + 25
     = 29.
"num: 0, rounded_num: 0": The third num is 0, and we round it to 0.
"sum_of_squares: 17": The rounded num is 0 and the square is 0**2 = 0.
    This step is wrong, so the sum of squares should be 4 + 25 + 0 = 29.
Finally, we compare the explanation of the test case with the log and the
     wrong result we got. We find the bug in the code: "rounded_num = int
    (num)" is wrong because from the log we found it doesn't round the
    number to its upper integer. We should round the input numbers to
    their upper integers. 1.4 should be rounded to 2 instead of 1, 4.2
    should be rounded to 5 instead of 4.
We should fix this bug according the above explanation:
**Code Start**
```
class Solution:
    def sum_squares(self, lst: List[int]) -> int:
        import math
        sum_of_squares = 0
        for num in lst:
            rounded_num = math.ceil(num)
            sum_of_squares += rounded_num**2
        return sum_of_squares
```
**Code End**
**Task End**

**Task Start**
**Description Start**
You are given two 0-indexed integer permutations A and B of length n.

A prefix common array of A and B is an array C such that C[i] is equal to
    the count of numbers that are present at or before the index i in
    both A and B.

Return the prefix common array of A and B.

A sequence of n integers is called apermutation if it contains all
    integers from 1 to n exactly once.
**Description End**
**Code Start**
```
class Solution:
    def findThePrefixCommonArray(self, A: List[int], B: List[int]) ->
    List[int]:
        common_array = []
        common_count = 0
        common_elements = set()

        for i in range(len(A)):
            if A[i] in B and B[i] in A:
                common_elements.add(A[i])
                common_elements.add(B[i])
            common_count = len(common_elements)
            common_array.append(common_count)
            print(f"i: {i}, A[i]: {A[i]}, B[i]: {B[i]}, common_elements:
    {common_elements}, common_count: {common_count}, common_array: {
    common_array}")

        print(f"return value: {common_array}")
        return common_array
```
**Code End**
Feedback: With the above function, we get findThePrefixCommonArray
    ([1,3,2,4], [3,1,2,4]) = [2, 2, 3, 4]. However, the test case is "
    findThePrefixCommonArray([1,3,2,4], [3,1,2,4]) == [0,2,3,4]". So the
    code doesn't pass the test case. Here's the log from the print()
    function in the code which can help debug:
**Log Start**
i: 0, A[i]: 1, B[i]: 3, common_elements: {1, 3}, common_count: 2,
    common_array: [2]
i: 1, A[i]: 3, B[i]: 1, common_elements: {1, 3}, common_count: 2,
    common_array: [2, 2]
i: 2, A[i]: 2, B[i]: 2, common_elements: {1, 2, 3}, common_count: 3,
    common_array: [2, 2, 3]
i: 3, A[i]: 4, B[i]: 4, common_elements: {1, 2, 3, 4}, common_count: 4,
    common_array: [2, 2, 3, 4]
return value: [2, 2, 3, 4]
**Log End**
The code above is wrong. Please find the bug by checking the test case
    and the log and fix it.
Explain both the test case and the log with the wrong answer, compare the
     explanation of the log to the explanation of the test case to find
    the bug:
### RAW COMPLETION
First, we explain the test case step by step. The test case is
    findThePrefixCommonArray([1,3,2,4], [3,1,2,4]) == [0,2,3,4] because:

At index 0, there are no common elements in A and B, so the common
array should have 0 at index 0. At index 1, A has elements [1, 3] and
 B has elements [3, 1], so there are 2 common elements (1 and 3), and
 the common array should have 2 at index 1. At index 2, A has
elements [1, 3, 2] and B has elements [3, 1, 2], so there are 3
common elements (1, 2, and 3), and the common array should have 3 at
index 2. At index 3, A has elements [1, 3, 2, 4] and B has elements
[3, 1, 2, 4], so there are 4 common elements (1, 2, 3, and 4), and
the common array should have 4 at index 3.

Second, we explain the log line by line and the wrong result we got from
    the wrong code:
"i: 0, A[i]: 1, B[i]: 3, common_elements: {1, 3}, common_count: 2,
    common_array: [2]": At index 0, A has element 1 and B has element 3.
    The code incorrectly adds both elements to the common_elements set,
    resulting in a common_count of 2. This is wrong because there are no
    common elements at index 0, so the common_count should be 0.
"i: 1, A[i]: 3, B[i]: 1, common_elements: {1, 3}, common_count: 2,
    common_array: [2, 2]": At index 1, A has elements [1, 3] and B has
    elements [3, 1]. The code correctly identifies the common elements (1
     and 3) and the common_count is 2. However, the common_array is wrong
     because it should be [0, 2] instead of [2, 2].
The rest of the log follows the same pattern, with the common_elements
    set being updated incorrectly at each step.

Finally, we compare the explanation of the test case with the log and the
     wrong result we got. We find the bug in the code: "if A[i] in B and
    B[i] in A:" is wrong because it checks if both A[i] and B[i] are in
    both A and B, which is not the correct condition for updating the
    common_elements set. We should update the common_elements set only
    when A[i] and B[i] are the same.

We should fix this bug according to the above explanation:
**Code Start**
```
class Solution:
    def findThePrefixCommonArray(self, A: List[int], B: List[int]) ->
    List[int]:
        common_array = []
        common_count = 0
        common_elements = set()

        for i in range(len(A)):
            if A[i] == B[i]:
                common_elements.add(A[i])
            common_count = len(common_elements)
            common_array.append(common_count)

        return common_array
```
**Code End**
### RESULTS
[Feedback from Leetcode online judging system]
wrong answer
Input:
[1,3,2,4]
[3,1,2,4]
Output:
[0,0,1,2]
Expected answer:
[0,2,3,4]

## ROUND 5
### PROMPT
**Task Start**
**Description Start**
You are given a list of numbers.
You need to return the sum of squared numbers in the given list, round
    each element in the list to the upper int(Ceiling) first.
class Solution:
    def sum_squares(self, lst: List[int]) -> int:
**Description End**
**Code Start**
class Solution:
    def sum_squares(self, lst: List[int]) -> int:
        sum_of_squares = 0
        for num in lst:
            rounded_num = int(num)
            sum_of_squares += rounded_num**2
        return sum_of_squares
**Code End**
Feedback: Don't change the code, but add some print() functions to print
    intermediate variables for debugging.
**Code Start**
class Solution:
    def sum_squares(self, lst: List[int]) -> int:
        sum_of_squares = 0
        for num in lst:
            rounded_num = int(num)
            print(f"num: {num}, rounded_num: {rounded_num}")
            sum_of_squares += rounded_num**2
            print(f"sum_of_squares: {sum_of_squares}")
        print(f"return value: {sum_of_squares}")
        return sum_of_squares
**Code End**
Feedback: With the above function, we get sum_squares([1.4,4.2,0]) = 17.
    However, the test case is "sum_squares([1.4,4.2,0]) == 29". So the
    code doesn't pass the test case. Here's the log from the print()
    function in the code which can help debug:
**Log Start**
num: 1.4, rounded_num: 1
sum_of_squares: 1
num: 4.2, rounded_num: 4
sum_of_squares: 17
num: 0, rounded_num: 0
sum_of_squares: 17
return value: 17
**Log End**
The code above is wrong. Please find the bug by checking the test case
    and the log and fix it.
Explain both the test case and the log with the wrong result, compare the
     explanation of the log to the explanation of the test case to find
    the bug:
First, we explain the test case step by step. The test case is
    sum_squares([1.4,4.2,0]) == 29 because: We round 1.4 to the upper
    integer 2 and the square of 2 is 2**2 = 4, we round 4.2 to the upper
    integer 5 and the square of 5 is 5**2 = 25, we round 0 to the upper
    integer 0 and the square of 0 is 0**2 = 0. The sum of squares are 4 +
     25 + 0 = 29, which is the same to the ground truth 29 in the test
    case "sum_squares([1.4,4.2,0]) == 29".

Second, we explain the log line by line and the wrong result we got from
    the wrong code:
"num: 1.4, rounded_num: 1": The first num of input is 1.4, and round it
    to 1. From here, we find the bug. It's wrong because we should round
    1.4 to its upper integer 2 instead of 1.
"sum_of_squares: 1": This step is wrong because the right rounded num
    should be 2. The square is 2**2 = 4, so the sum of squares is 4.
"num: 4.2, rounded_num: 4": The second num of input is 4.2, we round it
    to 4. From here, we find the bug. It's wrong because we should round
    4.2 to its upper integer 5 instead of 4.
"sum_of_squares: 17": This step is wrong because the right rounded num
    should be 5. The square is 5**2 = 25, so the sum of squares is 4 + 25
     = 29.
"num: 0, rounded_num: 0": The third num is 0, and we round it to 0.
"sum_of_squares: 17": The rounded num is 0 and the square is 0**2 = 0.
    This step is wrong, so the sum of squares should be 4 + 25 + 0 = 29.
Finally, we compare the explanation of the test case with the log and the
     wrong result we got. We find the bug in the code: "rounded_num = int
    (num)" is wrong because from the log we found it doesn't round the
    number to its upper integer. We should round the input numbers to
    their upper integers. 1.4 should be rounded to 2 instead of 1, 4.2
    should be rounded to 5 instead of 4.
We should fix this bug according the above explanation:
**Code Start**
class Solution:
    def sum_squares(self, lst: List[int]) -> int:
        import math
        sum_of_squares = 0
        for num in lst:
            rounded_num = math.ceil(num)
            sum_of_squares += rounded_num**2
        return sum_of_squares
**Code End**
**Task End**

**Task Start**
**Description Start**
You are given two 0-indexed integer permutations A and B of length n.

A prefix common array of A and B is an array C such that C[i] is equal to
     the count of numbers that are present at or before the index i in
    both A and B.

Return the prefix common array of A and B.

A sequence of n integers is called apermutation if it contains all
    integers from 1 to n exactly once.
**Description End**
**Code Start**
class Solution:
    def findThePrefixCommonArray(self, A: List[int], B: List[int]) ->
    List[int]:
        common_array = []
        common_count = 0
        common_elements = set()

        for i in range(len(A)):
            if A[i] == B[i]:
                common_elements.add(A[i])

```
        common_count = len(common_elements)
        common_array.append(common_count)
        print(f"i: {i}, A[i]: {A[i]}, B[i]: {B[i]}, common_elements:
    {common_elements}, common_count: {common_count}, common_array: {
    common_array}")

    print(f"return value: {common_array}")
    return common_array
```
**Code End**
Feedback: With the above function, we get findThePrefixCommonArray
    ([1,3,2,4], [3,1,2,4]) = [0, 0, 1, 2]. However, the test case is "
    findThePrefixCommonArray([1,3,2,4], [3,1,2,4]) == [0,2,3,4]". So the
    code doesn't pass the test case. Here's the log from the print()
    function in the code which can help debug:
**Log Start**
i: 0, A[i]: 1, B[i]: 3, common_elements: set(), common_count: 0,
    common_array: [0]
i: 1, A[i]: 3, B[i]: 1, common_elements: set(), common_count: 0,
    common_array: [0, 0]
i: 2, A[i]: 2, B[i]: 2, common_elements: {2}, common_count: 1,
    common_array: [0, 0, 1]
i: 3, A[i]: 4, B[i]: 4, common_elements: {2, 4}, common_count: 2,
    common_array: [0, 0, 1, 2]
return value: [0, 0, 1, 2]
**Log End**
The code above is wrong. Please find the bug by checking the test case
    and the log and fix it.
Explain both the test case and the log with the wrong answer, compare the
     explanation of the log to the explanation of the test case to find
    the bug:
### RAW COMPLETION
First, we explain the test case step by step. The test case is
    findThePrefixCommonArray([1,3,2,4], [3,1,2,4]) == [0,2,3,4] because:
    At index 0, there are no common elements in A and B, so the common
    array value is 0. At index 1, the common elements are 1 and 3, so the
     common array value is 2. At index 2, the common elements are 1, 3,
    and 2, so the common array value is 3. At index 3, all elements are
    common, so the common array value is 4. The final common array is [0,
     2, 3, 4].

Second, we explain the log line by line and the wrong result we got from
    the wrong code:
"i: 0, A[i]: 1, B[i]: 3, common_elements: set(), common_count: 0,
    common_array: [0]": The first element of A is 1 and the first element
     of B is 3. There are no common elements at index 0, so the common
    array value is 0. This step is correct.
"i: 1, A[i]: 3, B[i]: 1, common_elements: set(), common_count: 0,
    common_array: [0, 0]": The second element of A is 3 and the second
    element of B is 1. The code only checks if A[i] == B[i] and doesn't
    consider other elements before index i. This is the bug. The correct
    common array value should be 2 because elements 1 and 3 are common at
     index 1.
"i: 2, A[i]: 2, B[i]: 2, common_elements: {2}, common_count: 1,
    common_array: [0, 0, 1]": The third element of A is 2 and the third
    element of B is 2. The code only checks if A[i] == B[i] and doesn't
    consider other elements before index i. This is the bug. The correct
    common array value should be 3 because elements 1, 3, and 2 are
    common at index 2.

"i: 3, A[i]: 4, B[i]: 4, common_elements: {2, 4}, common_count: 2,
    common_array: [0, 0, 1, 2]": The fourth element of A is 4 and the
    fourth element of B is 4. The code only checks if A[i] == B[i] and
    doesn't consider other elements before index i. This is the bug. The
    correct common array value should be 4 because all elements are
    common at index 3.

Finally, we compare the explanation of the test case with the log and the
    wrong result we got. We find the bug in the code: "if A[i] == B[i]:"
    is wrong because from the log we found it doesn't consider other
    elements before index i. We should check if the elements in A and B
    are common at or before index i.

We should fix this bug according to the above explanation:
**Code Start**
```
class Solution:
    def findThePrefixCommonArray(self, A: List[int], B: List[int]) ->
    List[int]:
        common_array = []
        common_count = 0
        common_elements = set()

        for i in range(len(A)):
            if A[i] in B[:i+1]:
                common_elements.add(A[i])
            common_count = len(common_elements)
            common_array.append(common_count)

        return common_array
```
**Code End**
### RESULTS
[Feedback from Leetcode online judging system]
wrong answer
Input:
[1,3,2,4]
[3,1,2,4]
Output:
[0,1,2,3]
Expected answer:
[0,2,3,4]

## ROUND 6
### PROMPT
**Task Start**
**Description Start**
You are given a list of numbers.
You need to return the sum of squared numbers in the given list, round
    each element in the list to the upper int(Ceiling) first.
```
class Solution:
    def sum_squares(self, lst: List[int]) -> int:
```
**Description End**
**Code Start**
```
class Solution:
    def sum_squares(self, lst: List[int]) -> int:
        sum_of_squares = 0
        for num in lst:
            rounded_num = int(num)
            sum_of_squares += rounded_num**2
        return sum_of_squares
```

**Code End**
Feedback: Don't change the code, but add some print() functions to print
    intermediate variables for debugging.
**Code Start**
```
class Solution:
    def sum_squares(self, lst: List[int]) -> int:
        sum_of_squares = 0
        for num in lst:
            rounded_num = int(num)
            print(f"num: {num}, rounded_num: {rounded_num}")
            sum_of_squares += rounded_num**2
            print(f"sum_of_squares: {sum_of_squares}")
        print(f"return value: {sum_of_squares}")
        return sum_of_squares
```
**Code End**
Feedback: With the above function, we get sum_squares([1.4,4.2,0]) = 17.
    However, the test case is "sum_squares([1.4,4.2,0]) == 29". So the
    code doesn't pass the test case. Here's the log from the print()
    function in the code which can help debug:
**Log Start**
num: 1.4, rounded_num: 1
sum_of_squares: 1
num: 4.2, rounded_num: 4
sum_of_squares: 17
num: 0, rounded_num: 0
sum_of_squares: 17
return value: 17
**Log End**
The code above is wrong. Please find the bug by checking the test case
    and the log and fix it.
Explain both the test case and the log with the wrong result, compare the
     explanation of the log to the explanation of the test case to find
     the bug:
First, we explain the test case step by step. The test case is
    sum_squares([1.4,4.2,0]) == 29 because: We round 1.4 to the upper
    integer 2 and the square of 2 is 2**2 = 4, we round 4.2 to the upper
    integer 5 and the square of 5 is 5**2 = 25, we round 0 to the upper
    integer 0 and the square of 0 is 0**2 = 0. The sum of squares are 4 +
     25 + 0 = 29, which is the same to the ground truth 29 in the test
    case "sum_squares([1.4,4.2,0]) == 29".
Second, we explain the log line by line and the wrong result we got from
    the wrong code:
"num: 1.4, rounded_num: 1": The first num of input is 1.4, and round it
    to 1. From here, we find the bug. It's wrong because we should round
    1.4 to its upper integer 2 instead of 1.
"sum_of_squares: 1": This step is wrong because the right rounded num
    should be 2. The square is 2**2 = 4, so the sum of squares is 4.
"num: 4.2, rounded_num: 4": The second num of input is 4.2, we round it
    to 4. From here, we find the bug. It's wrong because we should round
    4.2 to its upper integer 5 instead of 4.
"sum_of_squares: 17": This step is wrong because the right rounded num
    should be 5. The square is 5**2 = 25, so the sum of squares is 4 + 25
     = 29.
"num: 0, rounded_num: 0": The third num is 0, and we round it to 0.
"sum_of_squares: 17": The rounded num is 0 and the square is 0**2 = 0.
    This step is wrong, so the sum of squares should be 4 + 25 + 0 = 29.
Finally, we compare the explanation of the test case with the log and the
     wrong result we got. We find the bug in the code: "rounded_num = int
    (num)" is wrong because from the log we found it doesn't round the

```
        number to its upper integer. We should round the input numbers to
        their upper integers. 1.4 should be rounded to 2 instead of 1, 4.2
        should be rounded to 5 instead of 4.
We should fix this bug according the above explanation:
**Code Start**
class Solution:
    def sum_squares(self, lst: List[int]) -> int:
        import math
        sum_of_squares = 0
        for num in lst:
            rounded_num = math.ceil(num)
            sum_of_squares += rounded_num**2
        return sum_of_squares
**Code End**
**Task End**
```

**Task Start**
**Description Start**
You are given two 0-indexed integer permutations A and B of length n.

A prefix common array of A and B is an array C such that C[i] is equal to
    the count of numbers that are present at or before the index i in
    both A and B.

Return the prefix common array of A and B.

A sequence of n integers is called apermutation if it contains all
    integers from 1 to n exactly once.
**Description End**
**Code Start**

```
class Solution:
    def findThePrefixCommonArray(self, A: List[int], B: List[int]) ->
    List[int]:
        common_array = []
        common_count = 0
        common_elements = set()

        for i in range(len(A)):
            if A[i] in B[:i+1]:
                common_elements.add(A[i])
            common_count = len(common_elements)
            common_array.append(common_count)
            print(f"i: {i}, A[i]: {A[i]}, B[:i+1]: {B[:i+1]},
    common_elements: {common_elements}, common_count: {common_count},
    common_array: {common_array}")

        print(f"return value: {common_array}")
        return common_array
```

**Code End**
Feedback: With the above function, we get findThePrefixCommonArray
    ([1,3,2,4], [3,1,2,4]) = [0, 1, 2, 3]. However, the test case is "
    findThePrefixCommonArray([1,3,2,4], [3,1,2,4]) == [0,2,3,4]". So the
    code doesn't pass the test case. Here's the log from the print()
    function in the code which can help debug:
**Log Start**
i: 0, A[i]: 1, B[:i+1]: [3], common_elements: set(), common_count: 0,
    common_array: [0]
i: 1, A[i]: 3, B[:i+1]: [3, 1], common_elements: {3}, common_count: 1,
    common_array: [0, 1]

```
i: 2, A[i]: 2, B[:i+1]: [3, 1, 2], common_elements: {2, 3}, common_count:
    2, common_array: [0, 1, 2]
i: 3, A[i]: 4, B[:i+1]: [3, 1, 2, 4], common_elements: {2, 3, 4},
    common_count: 3, common_array: [0, 1, 2, 3]
return value: [0, 1, 2, 3]
**Log End**
```
The code above is wrong. Please find the bug by checking the test case
    and the log and fix it.
Explain both the test case and the log with the wrong answer, compare the
    explanation of the log to the explanation of the test case to find
    the bug:
### RAW COMPLETION
First, we explain the test case step by step. The test case is
    findThePrefixCommonArray([1,3,2,4], [3,1,2,4]) == [0,2,3,4] because:
At index 0, there are no common elements in A and B, so the common array
    is [0].
At index 1, there are 2 common elements (1 and 3) in A[:2] and B[:2], so
    the common array is [0, 2].
At index 2, there are 3 common elements (1, 2, and 3) in A[:3] and B[:3],
    so the common array is [0, 2, 3].
At index 3, there are 4 common elements (1, 2, 3, and 4) in A[:4] and B
    [:4], so the common array is [0, 2, 3, 4].

Second, we explain the log line by line and the wrong result we got from
    the wrong code:
"i: 0, A[i]: 1, B[:i+1]: [3], common_elements: set(), common_count: 0,
    common_array: [0]": The first index is 0, and there are no common
    elements in A[:1] and B[:1], so the common array is [0].
"i: 1, A[i]: 3, B[:i+1]: [3, 1], common_elements: {3}, common_count: 1,
    common_array: [0, 1]": The second index is 1, and there are 2 common
    elements (1 and 3) in A[:2] and B[:2], but the code only finds 1
    common element (3), so the common array is [0, 1]. From here, we find
     the bug. It's wrong because we should find both common elements (1
    and 3) and the common array should be [0, 2].
"i: 2, A[i]: 2, B[:i+1]: [3, 1, 2], common_elements: {2, 3}, common_count
    : 2, common_array: [0, 1, 2]": The third index is 2, and there are 3
    common elements (1, 2, and 3) in A[:3] and B[:3], but the code only
    finds 2 common elements (2 and 3), so the common array is [0, 1, 2].
    This step is wrong, so the common array should be [0, 2, 3].
"i: 3, A[i]: 4, B[:i+1]: [3, 1, 2, 4], common_elements: {2, 3, 4},
    common_count: 3, common_array: [0, 1, 2, 3]": The fourth index is 3,
    and there are 4 common elements (1, 2, 3, and 4) in A[:4] and B[:4],
    but the code only finds 3 common elements (2, 3, and 4), so the
    common array is [0, 1, 2, 3]. This step is wrong, so the common array
     should be [0, 2, 3, 4].

Finally, we compare the explanation of the test case with the log and the
     wrong result we got. We find the bug in the code: "if A[i] in B[:i
    +1]:" is wrong because from the log we found it doesn't find all the
    common elements at or before the index i in both A and B. We should
    find all the common elements at or before the index i in both A and B
    .

We should fix this bug according to the above explanation:
**Code Start**
```
class Solution:
    def findThePrefixCommonArray(self, A: List[int], B: List[int]) ->
    List[int]:
        common_array = []
```

```
            common_count = 0
            common_elements = set()

            for i in range(len(A)):
                if A[i] in B[:i+1] and A[i] not in common_elements:
                    common_elements.add(A[i])
                    common_count += 1
                common_array.append(common_count)

            return common_array
```
**Code End**
**Task End**
### RESULTS
[Feedback from Leetcode online judging system]
wrong answer
Input:
[1,3,2,4]
[3,1,2,4]
Output:
[0,1,2,3]
Expected answer:
[0,2,3,4]

## ROUND 7
### PROMPT
**Task Start**
**Description Start**
You are given a list of numbers.
You need to return the sum of squared numbers in the given list, round
    each element in the list to the upper int(Ceiling) first.
class Solution:
    def sum_squares(self, lst: List[int]) -> int:
**Description End**
**Code Start**
```
class Solution:
    def sum_squares(self, lst: List[int]) -> int:
        sum_of_squares = 0
        for num in lst:
            rounded_num = int(num)
            sum_of_squares += rounded_num**2
        return sum_of_squares
```
**Code End**
Feedback: Don't change the code, but add some print() functions to print
    intermediate variables for debugging.
**Code Start**
```
class Solution:
    def sum_squares(self, lst: List[int]) -> int:
        sum_of_squares = 0
        for num in lst:
            rounded_num = int(num)
            print(f"num: {num}, rounded_num: {rounded_num}")
            sum_of_squares += rounded_num**2
            print(f"sum_of_squares: {sum_of_squares}")
        print(f"return value: {sum_of_squares}")
        return sum_of_squares
```
**Code End**
Feedback: With the above function, we get sum_squares([1.4,4.2,0]) = 17.
    However, the test case is "sum_squares([1.4,4.2,0]) == 29". So the

```
        code doesn't pass the test case. Here's the log from the print()
        function in the code which can help debug:
**Log Start**
num: 1.4, rounded_num: 1
sum_of_squares: 1
num: 4.2, rounded_num: 4
sum_of_squares: 17
num: 0, rounded_num: 0
sum_of_squares: 17
return value: 17
**Log End**
The code above is wrong. Please find the bug by checking the test case
        and the log and fix it.
Explain both the test case and the log with the wrong result, compare the
         explanation of the log to the explanation of the test case to find
         the bug:
First, we explain the test case step by step. The test case is
        sum_squares([1.4,4.2,0]) == 29 because: We round 1.4 to the upper
        integer 2 and the square of 2 is 2**2 = 4, we round 4.2 to the upper
        integer 5 and the square of 5 is 5**2 = 25, we round 0 to the upper
        integer 0 and the square of 0 is 0**2 = 0. The sum of squares are 4 +
         25 + 0 = 29, which is the same to the ground truth 29 in the test
        case "sum_squares([1.4,4.2,0]) == 29".
Second, we explain the log line by line and the wrong result we got from
        the wrong code:
"num: 1.4, rounded_num: 1": The first num of input is 1.4, and round it
        to 1. From here, we find the bug. It's wrong because we should round
        1.4 to its upper integer 2 instead of 1.
"sum_of_squares: 1": This step is wrong because the right rounded num
        should be 2. The square is 2**2 = 4, so the sum of squares is 4.
"num: 4.2, rounded_num: 4": The second num of input is 4.2, we round it
        to 4. From here, we find the bug. It's wrong because we should round
        4.2 to its upper integer 5 instead of 4.
"sum_of_squares: 17": This step is wrong because the right rounded num
        should be 5. The square is 5**2 = 25, so the sum of squares is 4 + 25
         = 29.
"num: 0, rounded_num: 0": The third num is 0, and we round it to 0.
"sum_of_squares: 17": The rounded num is 0 and the square is 0**2 = 0.
        This step is wrong, so the sum of squares should be 4 + 25 + 0 = 29.
Finally, we compare the explanation of the test case with the log and the
         wrong result we got. We find the bug in the code: "rounded_num = int
        (num)" is wrong because from the log we found it doesn't round the
        number to its upper integer. We should round the input numbers to
        their upper integers. 1.4 should be rounded to 2 instead of 1, 4.2
        should be rounded to 5 instead of 4.
We should fix this bug according the above explanation:
**Code Start**
class Solution:
    def sum_squares(self, lst: List[int]) -> int:
        import math
        sum_of_squares = 0
        for num in lst:
            rounded_num = math.ceil(num)
            sum_of_squares += rounded_num**2
        return sum_of_squares
**Code End**
**Task End**

**Task Start**
```

**Description Start**
You are given two 0-indexed integer permutations A and B of length n.

A prefix common array of A and B is an array C such that C[i] is equal to
    the count of numbers that are present at or before the index i in
    both A and B.

Return the prefix common array of A and B.

A sequence of n integers is called apermutation if it contains all
    integers from 1 to n exactly once.
**Description End**
**Code Start**
```
class Solution:
    def findThePrefixCommonArray(self, A: List[int], B: List[int]) ->
    List[int]:
        common_array = []
        common_count = 0
        common_elements = set()

        for i in range(len(A)):
            if A[i] in B[:i+1] and A[i] not in common_elements:
                common_elements.add(A[i])
                common_count += 1
            common_array.append(common_count)
            print(f"i: {i}, A[i]: {A[i]}, B[:i+1]: {B[:i+1]},
    common_elements: {common_elements}, common_count: {common_count},
    common_array: {common_array}")

        print(f"return value: {common_array}")
        return common_array
```
**Code End**
Feedback: With the above function, we get findThePrefixCommonArray
    ([1,3,2,4], [3,1,2,4]) = [0, 1, 2, 3]. However, the test case is "
    findThePrefixCommonArray([1,3,2,4], [3,1,2,4]) == [0,2,3,4]". So the
    code doesn't pass the test case. Here's the log from the print()
    function in the code which can help debug:
**Log Start**
i: 0, A[i]: 1, B[:i+1]: [3], common_elements: set(), common_count: 0,
    common_array: [0]
i: 1, A[i]: 3, B[:i+1]: [3, 1], common_elements: {3}, common_count: 1,
    common_array: [0, 1]
i: 2, A[i]: 2, B[:i+1]: [3, 1, 2], common_elements: {2, 3}, common_count:
     2, common_array: [0, 1, 2]
i: 3, A[i]: 4, B[:i+1]: [3, 1, 2, 4], common_elements: {2, 3, 4},
    common_count: 3, common_array: [0, 1, 2, 3]
return value: [0, 1, 2, 3]
**Log End**
The code above is wrong. Please find the bug by checking the test case
    and the log and fix it.
Explain both the test case and the log with the wrong answer, compare the
     explanation of the log to the explanation of the test case to find
    the bug:
### RAW COMPLETION
First, we explain the test case step by step. The test case is
    findThePrefixCommonArray([1,3,2,4], [3,1,2,4]) == [0,2,3,4] because:
At index 0, there are no common elements in A and B, so the common array
    should be [0].

At index 1, there are 2 common elements (1 and 3) in A[:2] and B[:2], so
    the common array should be [0, 2].
At index 2, there are 3 common elements (1, 2, and 3) in A[:3] and B[:3],
     so the common array should be [0, 2, 3].
At index 3, there are 4 common elements (1, 2, 3, and 4) in A[:4] and B
    [:4], so the common array should be [0, 2, 3, 4].

Second, we explain the log line by line and the wrong result we got from
    the wrong code:
"i: 0, A[i]: 1, B[:i+1]: [3], common_elements: set(), common_count: 0,
    common_array: [0]": The first index is 0, and there are no common
    elements in A[:1] and B[:1], so the common array is [0].
"i: 1, A[i]: 3, B[:i+1]: [3, 1], common_elements: {3}, common_count: 1,
    common_array: [0, 1]": The second index is 1, and there are 2 common
    elements (1 and 3) in A[:2] and B[:2]. However, the code only counts
    1 common element (3), so the common array is [0, 1] instead of [0,
    2]. This is where the bug is.
"i: 2, A[i]: 2, B[:i+1]: [3, 1, 2], common_elements: {2, 3}, common_count
    : 2, common_array: [0, 1, 2]": The third index is 2, and there are 3
    common elements (1, 2, and 3) in A[:3] and B[:3]. The code correctly
    counts 3 common elements, so the common array is [0, 1, 2, 3].
"i: 3, A[i]: 4, B[:i+1]: [3, 1, 2, 4], common_elements: {2, 3, 4},
    common_count: 3, common_array: [0, 1, 2, 3]": The fourth index is 3,
    and there are 4 common elements (1, 2, 3, and 4) in A[:4] and B[:4].
    However, the code only counts 3 common elements (2, 3, and 4), so the
     common array is [0, 1, 2, 3] instead of [0, 2, 3, 4].

Finally, we compare the explanation of the test case with the log and the
     wrong result we got. We find the bug in the code: "if A[i] in B[:i
    +1] and A[i] not in common_elements:" is wrong because from the log
    we found it doesn't count the common elements correctly. We should
    check if A[i] is in B[:i+1] and B[i] is in A[:i+1] instead.

We should fix this bug according to the above explanation:
**Code Start**
```python
class Solution:
    def findThePrefixCommonArray(self, A: List[int], B: List[int]) ->
    List[int]:
        common_array = []
        common_count = 0
        common_elements = set()

        for i in range(len(A)):
            if A[i] in B[:i+1] and B[i] in A[:i+1] and A[i] not in
    common_elements:
                common_elements.add(A[i])
                common_count += 1
            common_array.append(common_count)

        return common_array
```
**Code End**
**Task End**
### RESULTS
[Feedback from Leetcode online judging system]
wrong answer
Input:
[1,3,2,4]
[3,1,2,4]
Output:

```
[0,1,2,3]
Expected answer:
[0,2,3,4]
```

## ROUND 8
### PROMPT
**Task Start**
**Description Start**
You are given a list of numbers.
You need to return the sum of squared numbers in the given list, round
    each element in the list to the upper int(Ceiling) first.
class Solution:
    def sum_squares(self, lst: List[int]) -> int:
**Description End**
**Code Start**
```
class Solution:
    def sum_squares(self, lst: List[int]) -> int:
        sum_of_squares = 0
        for num in lst:
            rounded_num = int(num)
            sum_of_squares += rounded_num**2
        return sum_of_squares
```
**Code End**
Feedback: Don't change the code, but add some print() functions to print
    intermediate variables for debugging.
**Code Start**
```
class Solution:
    def sum_squares(self, lst: List[int]) -> int:
        sum_of_squares = 0
        for num in lst:
            rounded_num = int(num)
            print(f"num: {num}, rounded_num: {rounded_num}")
            sum_of_squares += rounded_num**2
            print(f"sum_of_squares: {sum_of_squares}")
        print(f"return value: {sum_of_squares}")
        return sum_of_squares
```
**Code End**
Feedback: With the above function, we get sum_squares([1.4,4.2,0]) = 17.
    However, the test case is "sum_squares([1.4,4.2,0]) == 29". So the
    code doesn't pass the test case. Here's the log from the print()
    function in the code which can help debug:
**Log Start**
num: 1.4, rounded_num: 1
sum_of_squares: 1
num: 4.2, rounded_num: 4
sum_of_squares: 17
num: 0, rounded_num: 0
sum_of_squares: 17
return value: 17
**Log End**
The code above is wrong. Please find the bug by checking the test case
    and the log and fix it.
Explain both the test case and the log with the wrong result, compare the
     explanation of the log to the explanation of the test case to find
    the bug:
First, we explain the test case step by step. The test case is
    sum_squares([1.4,4.2,0]) == 29 because: We round 1.4 to the upper
    integer 2 and the square of 2 is 2**2 = 4, we round 4.2 to the upper
    integer 5 and the square of 5 is 5**2 = 25, we round 0 to the upper

integer 0 and the square of 0 is 0**2 = 0. The sum of squares are 4 +
25 + 0 = 29, which is the same to the ground truth 29 in the test
case "sum_squares([1.4,4.2,0]) == 29".
Second, we explain the log line by line and the wrong result we got from
the wrong code:
"num: 1.4, rounded_num: 1": The first num of input is 1.4, and round it
to 1. From here, we find the bug. It's wrong because we should round
1.4 to its upper integer 2 instead of 1.
"sum_of_squares: 1": This step is wrong because the right rounded num
should be 2. The square is 2**2 = 4, so the sum of squares is 4.
"num: 4.2, rounded_num: 4": The second num of input is 4.2, we round it
to 4. From here, we find the bug. It's wrong because we should round
4.2 to its upper integer 5 instead of 4.
"sum_of_squares: 17": This step is wrong because the right rounded num
should be 5. The square is 5**2 = 25, so the sum of squares is 4 + 25
= 29.
"num: 0, rounded_num: 0": The third num is 0, and we round it to 0.
"sum_of_squares: 17": The rounded num is 0 and the square is 0**2 = 0.
This step is wrong, so the sum of squares should be 4 + 25 + 0 = 29.
Finally, we compare the explanation of the test case with the log and the
wrong result we got. We find the bug in the code: "rounded_num = int
(num)" is wrong because from the log we found it doesn't round the
number to its upper integer. We should round the input numbers to
their upper integers. 1.4 should be rounded to 2 instead of 1, 4.2
should be rounded to 5 instead of 4.
We should fix this bug according the above explanation:
**Code Start**
class Solution:
    def sum_squares(self, lst: List[int]) -> int:
        import math
        sum_of_squares = 0
        for num in lst:
            rounded_num = math.ceil(num)
            sum_of_squares += rounded_num**2
        return sum_of_squares
**Code End**
**Task End**

**Task Start**
**Description Start**
You are given two 0-indexed integer permutations A and B of length n.

A prefix common array of A and B is an array C such that C[i] is equal to
the count of numbers that are present at or before the index i in
both A and B.

Return the prefix common array of A and B.

A sequence of n integers is called apermutation if it contains all
integers from 1 to n exactly once.
**Description End**
**Code Start**
class Solution:
    def findThePrefixCommonArray(self, A: List[int], B: List[int]) ->
    List[int]:
        common_array = []
        common_count = 0
        common_elements = set()

```
    for i in range(len(A)):
        print(f"i: {i}, A[i]: {A[i]}, B[i]: {B[i]}, common_elements:
{common_elements}")
        if A[i] in B[:i+1] and B[i] in A[:i+1] and A[i] not in
common_elements:
            common_elements.add(A[i])
            common_count += 1
        common_array.append(common_count)
        print(f"common_array: {common_array}")

    return common_array
```
**Code End**
Feedback: With the above function, we get findThePrefixCommonArray
    ([1,3,2,4], [3,1,2,4]) = [0, 1, 2, 3]. However, the test case is "
    findThePrefixCommonArray([1,3,2,4], [3,1,2,4]) == [0,2,3,4]". So the
    code doesn't pass the test case. Here's the log from the print()
    function in the code which can help debug:
**Log Start**
i: 0, A[i]: 1, B[i]: 3, common_elements: set()
common_array: [0]
i: 1, A[i]: 3, B[i]: 1, common_elements: set()
common_array: [0, 1]
i: 2, A[i]: 2, B[i]: 2, common_elements: {3}
common_array: [0, 1, 2]
i: 3, A[i]: 4, B[i]: 4, common_elements: {2, 3}
common_array: [0, 1, 2, 3]
**Log End**
The code above is wrong. Please find the bug by checking the test case
    and the log and fix it.
Explain both the test case and the log with the wrong answer, compare the
    explanation of the log to the explanation of the test case to find
    the bug:
### RAW COMPLETION
First, we explain the test case step by step. The test case is
    findThePrefixCommonArray([1,3,2,4], [3,1,2,4]) == [0,2,3,4] because:
At index 0, there are no common elements in A and B, so the common array
    value is 0.
At index 1, both A and B have elements 1 and 3, so the common array value
    is 2.
At index 2, both A and B have elements 1, 3, and 2, so the common array
    value is 3.
At index 3, both A and B have elements 1, 3, 2, and 4, so the common
    array value is 4.
The prefix common array is [0, 2, 3, 4].

Second, we explain the log line by line and the wrong result we got from
    the wrong code:
"i: 0, A[i]: 1, B[i]: 3, common_elements: set()": The first index is 0, A
    [0] is 1, B[0] is 3, and there are no common elements yet.
"common_array: [0]": At index 0, there are no common elements, so the
    common array value is 0.

"i: 1, A[i]: 3, B[i]: 1, common_elements: set()": The second index is 1,
    A[1] is 3, B[1] is 1, and there are no common elements yet.
"common_array: [0, 1]": This step is wrong because both A and B have
    elements 1 and 3 at or before index 1, so the common array value
    should be 2.

"i: 2, A[i]: 2, B[i]: 2, common_elements: {3}": The third index is 2, A
    [2] is 2, B[2] is 2, and the common elements are {3}.
"common_array: [0, 1, 2]": At index 2, both A and B have elements 1, 3,
    and 2, so the common array value is 3.

"i: 3, A[i]: 4, B[i]: 4, common_elements: {2, 3}": The fourth index is 3,
     A[3] is 4, B[3] is 4, and the common elements are {2, 3}.
"common_array: [0, 1, 2, 3]": This step is wrong because both A and B
    have elements 1, 3, 2, and 4 at or before index 3, so the common
    array value should be 4.

Finally, we compare the explanation of the test case with the log and the
     wrong result we got. We find the bug in the code: "if A[i] in B[:i
    +1] and B[i] in A[:i+1] and A[i] not in common_elements" is wrong
    because it only checks if A[i] and B[i] are common elements, but it
    should check all elements at or before index i in both A and B.

We should fix this bug according to the above explanation:

**Code Start**
```
class Solution:
    def findThePrefixCommonArray(self, A: List[int], B: List[int]) ->
    List[int]:
        common_array = []
        common_count = 0
        common_elements = set()

        for i in range(len(A)):
            if A[i] not in common_elements and A[i] in B[:i+1]:
                common_elements.add(A[i])
                common_count += 1
            if B[i] not in common_elements and B[i] in A[:i+1]:
                common_elements.add(B[i])
                common_count += 1
            common_array.append(common_count)

        return common_array
```
**Code End**
### RESULTS
[Feedback from Leetcode online judging system]
succeed

