# OpenReview forum: "Leveraging Print Debugging to Improve Code Generation in Large Language Models"
_ICLR.cc/2024/Conference — ICLR 2024 Conference Withdrawn Submission_

### Official Review · Reviewer_afP2 · 2023-10-27

**Soundness:** 3 good
**Presentation:** 3 good
**Contribution:** 2 fair
**Rating:** 5
**Confidence:** 5

**Summary:**

While LLMs have improved strongly on program generation, they struggle with solving hard, algorithmic problems. A variety of techniques have been proposed to allow LLMs to debug their own outputs, e.g. based on test feedback, in order to improve their responses gradually and solve more tasks. This work proposes one such method, based on prompting an LLM (GPT-4) to generate print-line debugging statements in its solution. The model is then provided with the output of running the code and tasked with improving the code. Experiments show that this allows a model to gradually improve over many steps, ultimately solving substantially more medium-difficulty coding problems than prior work.

**Strengths:**

The main value of this work lies in the strong performance improvement it shows on medium-difficulty programming problems, where it nearly doubles the fraction of problems solved compared to prior work. In particular, the technique shows potential in continuing to solve more problems over the course of repeated iterations. Both of these results are quite significant.

The approach itself is relatively straightforward. It sits at the intersection of basic prompting strategies, learning from execution feedback, and tool usage. The paper was largely fairly easy to follow.

**Weaknesses:**

The contribution is very slim. The work offers no real theoretical or conceptual contributions. The approach consists of prompting an LLM and feeding back the result of the program's execution. The benefit of this approach is demonstrated on a relatively narrow set of problems (mainly, medium-level programming challenges). The work also involves fairly few ablations and analyses of alternatives. As such, the contribution largely lies in the choice of prompt. The "Questions" section below offers a range of ideas for expanding the investigation to make the contribution more substantial and complete.

**Questions:**

My main impression of this work is that it establishes a prompting strategy that works for a very specific set of problems. While it works quite well on those, I would like to see it explore this domain more broadly. Please consider and respond to questions like:
- Is the LLM very sensitive to the wording of the prompts? Is it sensitive to the placement of the print statements? Is there evidence that it is especially capable at picking printing locations that will maximize its odds of success, or could a heuristic baseline be established that would pick similarly effective print statements?
- What types of training signal are (likely) required on the LLM's end to leverage this type of feedback effectively. Are there implications for training future LLMs based on these insights? Why were no other LLMs investigated?
- Why do you believe that unit test feedback is less useful as a training signal, in particular after the first step (Fig. 4)? What experiments might be conducted to identify in more detail why this technique does not work at all on hard problems, and where exactly the difference between easy and hard problems lies that makes it so that no technique works well on the latter? Could a research direction inspired by tihs work unlock the type of advanced capabilities required to solve harder problems, and if so, how?
- Do you expect a form of this idea to translate to other communities, like NLP tasks? One framing of this approach is one of tool usage, where an inspection tool is invoked by the LLM. In that framing, here are certainly counterparts in other domains, such as a dictionary lookup, a web search query, or a simulation. At the same time, tool usage has already been widely explored. How do you position the conceptual contribution of this work in that light?

Minor comments:
- Tab. 1: consider expanding or upper-casing "ut"
- Results: consider stipulating that these are absolute percentage points, not relative percentages. When I initially read 17.9%, I expected a much less substantial improvement that it turned out to be.
- P7: "in 2." -> "in Table 2."
- Fig. 4: it's a bit surprising that all the other techniques immediately saturate after one step. I would have expected unit test feedback, for instance, to have at least a somewhat similar curve to the print debugging approach. Please consider double-checking the experimental setup used here.

---

### Official Review · Reviewer_VJBR · 2023-10-28

**Soundness:** 3 good
**Presentation:** 2 fair
**Contribution:** 2 fair
**Rating:** 3
**Confidence:** 3

**Summary:**

This work is about automated program generation and repair. The proposed method improves another LLM (large language model) based rubber duck debugging work, by inserting print statements into the code. The LLM will decide the locations and the number of statements to add by itself. These print statements aim to capture the changing state and generate logs for debugging purposes. Intuitively, this is similar to how human programmers debug failed test case.

**Strengths:**

The proposed method is simple and effective. It outperforms the existing work on selected dataset.

**Weaknesses:**

Without thorough explanation and analysis of its efficacy, the proposed method appears to be an incremental extension of rubber duck debugging, leaving it at risk of being overshadowed by alternative strategies for fine-tuning or tweaking the use of Large Language Models (LLMs) in program generation and repair.

The baselines used in the comparison do not represent the state of the art. There is a large number of automated program repair techniques, including many using LLMs, and they are not included.  The selected dataset does not seem  to be comprehensive or diverse, which also weakens the results. Furthermore, the experiments for the proposed method exclusively utilise the GPT-4 model, casting doubt on the generality and applicability of the proposed print-statement for debugging.

The experimental results indicate that only a few print statements are needed in the proposed debugging method, which is interesting. However, this outcome may also be dependent on the dataset used.

**Questions:**

1. How can the superior performance of the proposed method compared to rubber duck debugging be explained and justified?

2. How does the proposed method relate to the automated program repair methods by large language models?

3. Is the proposed method expected to remain effective when applied to more general and diverse datasets, as well as with various other large language models?

---

### Official Review · Reviewer_Yd6T · 2023-10-31

**Soundness:** 2 fair
**Presentation:** 3 good
**Contribution:** 2 fair
**Rating:** 3
**Confidence:** 5

**Summary:**

This paper introduces a prompting approach to steer large language models (LLMs) towards debugging code using the "print debugging" method.

**Strengths:**

The idea of leveraging print debugging for LLMs is straightforward and well-motivated. Print debugging is an intuitive technique used by human programmers, so teaching this to LLMs could improve their debugging abilities.

**Weaknesses:**

While the suggested method employs a practical prompting strategy, it falls short in comparisons on two fronts: 1. across multiple datasets and 2. with diverse CodeLLM baselines.
* Regarding datasets: The rubber duck paper assessed its methodology across a variety of readily available datasets that come with unit tests, and easy-to-integrate interpreters. This paper should broaden its scope by evaluating the prompting method on more tasks and datasets, such as TransCoder (with 5 unit tests), MBPP (with 3 unit tests), and Spider.
* Regarding CodeLLMs: The GPT-4 webpage version boasts data analysis capabilities and can automatically debug itself through error logs. One could guess that the close-source GPT-4 has been fine-tuned for self-correction based on logs. Thus, it is imperative for this study to assess the print prompting technique on other open-source CodeLLMs like CodeLLAMA. A side-by-side evaluation (e.g., behavior differences) of various CodeLLMs utilizing the print prompting method would also bring more insights to future work.

**Questions:**

* An error analysis on the baseline model would be beneficial. e.g., what is the percentage of each bug type that can be identified using print statements (e.g., out of bound, value error, key error, wrong algorithm, syntax error)?
* In section 4.1, why is the max token count set to 4096 when employing a 32k model?

Missing references on LLM Code Tracing
* Code Execution with Pre-trained Language Models
* GRAPHCODEBERT: PRE-TRAINING CODE REPRESENTATIONS WITH DATA FLOW

---

### Official Review · Reviewer_aiKM · 2023-10-31

**Soundness:** 3 good
**Presentation:** 3 good
**Contribution:** 3 good
**Rating:** 6
**Confidence:** 4

**Summary:**

Authors propose to use print debugging to improve LLM based code generation. They use Leetcode problem dataset and demonstrate that print debugging approach outperforms previous rubber duck debugging approach.

**Strengths:**

- Propose print debugging to improve code generation
- Demonstrate significant outperformance vs. recent "rubber duck debugging" approach on medium-level Leetcode problems
- Show that debugging methods can improve easy and medium problem solutions, but cannot improve hard problem solutions that probably require deeper algorithmic, structural, or semantic understanding
- Leetcode problems dataset

**Weaknesses:**

- Experiments performed only with GPT-4 which is a good candidate LLM but only a single candidate.

**Questions:**

No questions